# Simple Stochastic and Online Gradient Descent Algorithms for Pairwise Learning

**Zhenhuan Yang**[1*]    **Yunwen Lei**[2*]    **Puyu Wang**[3]    **Tianbao Yang**[4]    **Yiming Ying**[1]

[1]University at Albany, SUNY, Albany, NY    [2]University of Birmingham, Birmingham
[3]City University of Hong Kong, Hong Kong    [4]University of Iowa City, IA
zyang6@albany.edu, y.lei@hbam.ac.uk, puyuwang@cityu.edu.hk,
tianbao-yang@uiowa.edu, yying@albany.edu

## Abstract

Pairwise learning refers to learning tasks where the loss function depends on a pair of instances. It instantiates many important machine learning tasks such as bipartite ranking and metric learning. A popular approach to handle streaming data in pairwise learning is an online gradient descent (OGD) algorithm, where one needs to pair the current instance with a buffering set of previous instances with a sufficiently large size and therefore suffers from a scalability issue. In this paper, we propose simple stochastic and online gradient descent methods for pairwise learning. A notable difference from the existing studies is that we only pair the current instance with the previous one in building a gradient direction, which is efficient in both the storage and computational complexity. We develop novel stability results, optimization, and generalization error bounds for both convex and nonconvex as well as both smooth and nonsmooth problems. We introduce novel techniques to decouple the dependency of models and the previous instance in both the optimization and generalization analysis. Our study resolves an open question on developing meaningful generalization bounds for OGD using a buffering set with a very small fixed size. We also extend our algorithms and stability analysis to develop differentially private SGD algorithms for pairwise learning which significantly improves the existing results.

## 1 Introduction

Many important learning tasks involve pairwise loss functions which are often referred to as pairwise learning. Such notable learning tasks include AUC maximization [15, 22, 28, 37, 46, 48], metric learning [4, 24, 39, 41, 44], and a minimum error entropy principle [19]. For instance, AUC maximization aims to rank positive instances above negative ones which involves a loss $f(\mathbf{w}; (x, y), (x', y')) = (1 - \mathbf{w}^\top(x - x'))_+ \mathbb{I}_{[y=1 \wedge y'=-1]}$ with $x, x' \in \mathcal{X} \subseteq \mathbb{R}^d$ and $y, y' \in \mathcal{Y} = \{\pm 1\}$. The aim of metric learning is to learn a distance function $h_{\mathbf{w}}(x, x') = (x - x')^\top \mathbf{w}(x - x')$ where $\mathbf{w}$ is positive semi-definite matrix in $\mathbb{R}^{d \times d}$. A typical pairwise loss can be $f(\mathbf{w}; (x, y), (x', y')) = (1 + \tau(y, y')h_{\mathbf{w}}(x, x'))_+$ where $\tau(y, y') = 1$ if $y = y'$ and $-1$ otherwise. Given a training data $S = \{z_i = (x_i, y_i) \in \mathcal{X} \times \mathcal{Y} : i \in [n]\}$ where $[n] = \{1, 2, \ldots, n\}$, the ERM formulation for pairwise learning is defined as $\min_{\mathbf{w} \in \mathcal{W}} \frac{1}{n(n-1)} \sum_{i,j \in [n], i \neq j}^{n} f(\mathbf{w}; z_i, z_j)$ where $\mathcal{W}$ denotes the parameter space. This scheme has been well studied theoretically using algorithmic stability [1] and U-statistics tools [11]. At the same time, there are considerable interests on developing and studying online gradient descent (OGD) or stochastic gradient descent (SGD) algorithms for pairwise learning due to their scalability in practice.

---

*Equal contributions. The corresponding author is Yiming Ying.

The critical issue for designing such stochastic algorithms is to construct intersecting pairs of instances for updating the model parameter upon receiving individual instances. For the offline (finite-sum) setting where the prescribed training data $S = \{z_1, \ldots, z_n\}$ is available, one natural approach is to, at each time $t$, randomly select a pair of instances $(z_{i_t}, z_{j_t})$ from $n(n-1)/2$ pairs and update $\mathbf{w}$ based on the gradient of the local error $f(\mathbf{w}_{t-1}; z_{i_t}, z_{j_t})$. The excess generalization and stability was nicely established in [27]. One popular approach [22, 38, 45, 48] is to consider the online setting where the data is continuously arriving. This approach pairs the current datum $z_t = (x_t, y_t)$, which is received at time $t$, with all previous instances $S_{t-1} = \{z_1, \ldots, z_{t-1}\}$ and then performs the update based on the gradient of the local error $f(\mathbf{w}_{t-1}; S_{t-1}) = \frac{1}{t-1} \sum_{z \in S_{t-1}} f(\mathbf{w}_{t-1}; z_t, z)$. It requires a high gradient complexity $\mathcal{O}(t)$ (i.e. the number of computing gradients) which is expensive when $t$ becomes large. To mitigate this potential limitation, [22, 37, 48] proposed to use a buffering set $B_{t-1} \subseteq S_{t-1}$ of size $s$ and the local error $f(\mathbf{w}_{t-1}; B_{t-1}) = \frac{1}{s} \sum_{z \in B_{t-1}} f(\mathbf{w}_{t-1}; z_t, z)$ which reduces the gradient complexity to $\mathcal{O}(s)$. The excess generalization bound $\mathcal{O}(\frac{1}{\sqrt{s}} + \frac{1}{\sqrt{n}})$ was established in [22] using the online-to-batch conversion method [8] which is only meaningful for a very large $s$. In particular, this bound tends to zero only when $s = s(n)$ tends to infinity as $n$ tends to infinity. It was mentioned in [22] (see the discussion at the end of Section 7 there) as an open question on how to get a meaningful bound for a fixed constant $s$.

In this paper, we show that optimal generalization bounds can be achieved for simple SGD and OGD algorithms for pairwise learning where, at time $t$, the current instance $z_t$ is only paired with the previous instance $z_{t-1}$. This is equivalent to the First-In-First-Out (FIFO) buffering strategy [22, 37] while keeping the size $s$ of the buffering set $B_{t-1}$ to be $s = 1$, where, in this FIFO policy, the data $z_t$ arriving at time $t > 1$ is included into the buffer by removing $\{z_1, \ldots, z_{t-2}\}$ from the buffer. In particular, our main contributions are summarized as follows.

- We propose simple SGD and OGD algorithms for pairwise learning where the $t$-th update of the model parameter is based on the interacting of the current instance and the previous one which has a constant gradient complexity $\mathcal{O}(1)$.

- We establish the stability results of the proposed SGD algorithms for pairwise learning and apply them to derive optimal excess generalization bounds $\mathcal{O}(1/\sqrt{n})$ for the proposed simple SGD algorithm for pairwise learning with both convex and nonconvex as well as both smooth and nonsmooth losses in the offline (finite-sum) setting where the training data of size $n$ is given. We introduce novel techniques to decouple the dependency of the current SGD iterate with the previous instance in both the generalization and optimization error analysis, which resolves the open question in [22] on how to develop meaningful generalization bounds when the buffering set of FIFO has a very small size.

- We further develop a localization version of our SGD algorithms under $(\epsilon, \delta)$-differential privacy (DP) constraints, and apply the obtained stability results to derive an optimal utility (excess generalization) bound $\tilde{\mathcal{O}}(1/\sqrt{n} + \sqrt{d}/n\epsilon)$. In contrast to the existing work [21] which requires the loss function to be smooth and an at least quadratic gradient complexity, our proposed DP algorithms only need a linear gradient complexity $\tilde{\mathcal{O}}(n)$ for smooth convex losses to achieve the optimal utility bound and can also be applied to non-smooth convex losses.

The paper is organized as follows. Section 2 reviews the related work and Section 3 describes the proposed algorithms. In Section 4, we present excess generalization bounds, stability results, and optimization errors of our algorithms. Section 5 is devoted to the differentially private SGD for pairwise learning and its utility bounds. Section 6 provides experimental validation of theoretical findings. The paper is concluded in Section 7. All the main proofs are postponed to the Appendix.

## 2 Motivating Examples and Related Work

In this section, we list examples of pairwise learning and discuss some related work.

### 2.1 Motivating Examples

**AUC maximization.** Area under the ROC curve (AUC) of a prediction function $h_{\mathbf{w}}$ is the probability that the function ranks a random positive example higher than a random negative example. The empirical risk of AUC maximization is given by $F_S(\mathbf{w}) = \frac{1}{n(n-1)} \sum_{i,j \in [n], i \neq j} \ell(h_{\mathbf{w}}(\mathbf{x}_i) -$

$h_{\mathbf{w}}(\mathbf{x}_j))\mathbb{I}_{[y_i=1]}\mathbb{I}_{[y_j=-1]}$, where the loss $\ell(\cdot)$ can be the least square loss $\ell(t) = (1-t)^2$ or the hinge loss $\ell(t) = (1-t)_+$.

**Minimum error entropy principle.** Minimum error entropy (MEE) is a principle of information theoretical learning [19, 20], which aims to find a predictor $h : \mathcal{X} \mapsto \mathcal{Y}$ by minimizing the information entropy of the variable $E = Y - h(X)$. The Rényi's entropy of order 2 for $E$ is defined as $H(E) = -\log \int p_E^2(e)de$. Here $p_E$ is probability density function and can be approximated by Parzen windowing $\hat{p}_E(e) = \frac{1}{n\gamma} \sum_{i=1}^{n} G\big(\frac{(e-e_i)^2}{2\gamma^2}\big)$, where $e_i = y_i - h(x_i)$, and $\gamma > 0$ is an MEE scaling parameter and $G : \mathbb{R} \mapsto \mathbb{R}^+$ is a windowing function. The approximation of Rényi's entropy is given by its empirical version $\hat{H} = -\log \frac{1}{n^2\gamma} \sum_{i,j\in[n]} G\big(\frac{(e_i-e_j)^2}{2\gamma^2}\big)$. The maximization of $\hat{H}$ then leads to a pairwise learning problem since each loss function involves a pair of training examples [19].

**Metric learning.** Metric learning aims to fine a distance metric $h_{\mathbf{w}} : \mathcal{X} \times \mathcal{X} \mapsto \mathbb{R}_+$ consistent with some supervised information, e.g., examples within the same class are close while examples from different classes are apart from each other under the learnt metric. If $\mathcal{Y} = \{\pm 1\}$, the performance of $h$ on an example pair $z, z'$ can be quantified by a loss function of the form $f(\mathbf{w}; z, z') = \ell(yy'(1 - h_{\mathbf{w}}; x, x'))$, where $\ell : \mathbb{R} \mapsto \mathbb{R}_+$ is a decreasing function for which some typical choices include the hinge loss $\ell(t) = (1-t)_+$ and the exponential function $\ell(t) = \log(1 + \exp(-t))$. Then one can minimize the training error $F_S(\mathbf{w}) = \frac{1}{n(n-1)} \sum_{i,j\in[n]:i\neq j} f(\mathbf{w}; z_i, z_j)$ to learn a distance metric. Another closely related learning task to metric learning is constrastive learning [10, 30, 35, 40] which has become very popular recently for learning visual representations without supervision.

## 2.2 Related Work

The work [22, 37, 38, 48] assumed the online learning setting in which a stream of i.i.d. data $\{z_1, z_2, \ldots, z_t, \ldots\}$ is continuously arriving. Upon receiving $z_t$ at time $t$, it is paired with all previous instances and then the model parameter is updated based on the local error $F_t(\mathbf{w}_{t-1}) = \frac{1}{t-1} \sum_{j=1}^{t-1} f(\mathbf{w}_{t-1}; \mathbf{z}_t, \mathbf{z}_j)$. In particular, the work [37, 38] provided the first excess generalization bound for online learning methods by obtaining online-to-batch conversion bounds [8] using covering numbers of function classes. Kar et al. [22] significantly improved the results using the so-called symmetrization of expectations which reduce excess risk estimates to Rademacher complexities. To further reduce the expensive gradient complexity $\mathcal{O}(t)$ at a large time $t$, the work [22, 38, 48] proposed to use a buffering set $B_{t-1}$ with size $s$ instead of all previous instances. It was shown that such OGD algorithms have an excess generalization bound $\mathcal{O}(\frac{1}{\sqrt{s}} + \frac{1}{\sqrt{n}})$ for convex and Lipschitz-continuous losses. In the offline learning setting where the training data $S = \{z_1, z_2, \ldots, z_n\}$ of size $n$ is fixed, the work [43] considered the stochastic version of the algorithm in [22, 37] where, at time $t$, a random instance $z_{i_t}$ with random index $i_t \in \{1, \ldots, n\}$ is paired with all previous instances $\{z_{i_1}, \ldots, z_{i_{t-1}}\}$. They derived stability and generalization results of such algorithms in expectation.

Lei et al. [27] considered the offline learning setting and, at time $t$, the algorithm there randomly picks a pair of instances $(\mathbf{z}_{i_t}, \mathbf{z}_{j_t})$ from all $\binom{n}{2}$ pairs of instances. An excess risk bound $\tilde{\mathcal{O}}(1/\sqrt{n})$ with high probability was derived for convex, Lipschitz and strongly smooth losses. Here the notation $\tilde{\mathcal{O}}(\cdot)$ means $\mathcal{O}(\cdot)$ up to some logarithmic terms. In the particular case of AUC maximization with the least square loss, [46] considered the online learning setting and reformulated the problem as a stochastic saddle point (min-max) problem which decouples the pairwise structure. From this reformulation, efficient SGD-type algorithms [29, 44] have been developed.

Recently, differentially private pairwise learning has been studied where the construction of pairs of instances follows [22, 37, 48] or all pairs are used at each iteration. In particular, Huai et al. [21] considered both online and offline learning settings and the pairs of instances at time $t$ follow [22, 37, 48]. They provided a utility bound $\widetilde{\mathcal{O}}(\sqrt{d}/(\sqrt{n}\epsilon))$ for convex and smooth loss functions. The study [43] also paired the current instance with all previous ones and showed that SGD with output perturbation for pairwise learning has a utility bound $\widetilde{\mathcal{O}}(\sqrt{d}/(\sqrt{n}\epsilon))$ for nonsmooth convex losses. The work [42] showed that private gradient descent using all possible pairs can achieve a utility bound $\widetilde{\mathcal{O}}(1/\sqrt{n} + \sqrt{d}/(n\epsilon))$ for strongly smooth and convex losses.

# 3 Proposed Algorithms for Pairwise Learning

In this section, we describe the proposed algorithms in two common learning settings for pairwise learning: offline and online learning settings. Let $\rho$ be a probability measure defined on $\mathcal{Z} := \mathcal{X} \times \mathcal{Y}$, where $\mathcal{X}$ is an input space and $\mathcal{Y}$ is an output space. In pairwise learning, the performance of $\mathbf{w}$ is measured on a pair of instances $(z, z')$ by a nonnegative loss function $f(\mathbf{w}; z, z')$. Denote by $[n] := \{1, 2, \ldots, n\}$ for any $n \in \mathbb{N}$.

| **Algorithm 1** SGD for Pairwise Learning | **Algorithm 2** OGD for Pairwise Learning |
|---|---|
| 1: **Inputs:** $S = \{z_i : i \in [n]\}$ and step sizes $\{\eta_t\}$ | 1: **Inputs:** learning rates $\{\eta_t\}$ |
| 2: **Initialize:** $\mathbf{w}_0 \in \mathcal{W}$, let $\mathbf{w}_{-1} = \mathbf{w}_0$ and randomly select $i_0 \in [n]$ | 2: **Initialize:** $\mathbf{w}_0 \in \mathcal{W}$, let $\mathbf{w}_{-1} = \mathbf{w}_0$ and receiving datum $z_0$. |
| 3: **for** $t = 1, 2, \ldots, T$ **do** | 3: **for** $t = 1, 2, \ldots, T$ **do** |
| 4:    Randomly select $i_t \in [n]$ | 4:    Receive a data point $z_t$ |
| 5:    $\mathbf{w}_t = \Pi_{\mathcal{W}}\big(\mathbf{w}_{t-1} - \eta_t \nabla f(\mathbf{w}_{t-1}; z_{i_t}, z_{i_{t-1}})\big)$ | 5:    $\mathbf{w}_t = \Pi_{\mathcal{W}}\big(\mathbf{w}_{t-1} - \eta_t \nabla f(\mathbf{w}_{t-1}; z_t, z_{t-1})\big)$ |
| 6: **Outputs:** $\bar{\mathbf{w}}_T = \sum_{j=1}^{T} \eta_j \mathbf{w}_{j-2} / \sum_{j=1}^{T} \eta_j$ | 6: **Outputs:** $\widetilde{\mathbf{w}}_T = \sum_{j=1}^{T} \eta_j \mathbf{w}_{j-2} / \sum_{j=1}^{T} \eta_j$ |

**Offline Learning (Finite-Sum) Setting.** The first is the finite-sum setting where the training data $S = \{z_i = (x_i, y_i) \in \mathcal{Z} : i \in [n]\}$ are drawn independently according to $\rho$. In this context, one aims to solve the following empirical risk minimization (ERM):

$$\mathbf{w}_S^* = \underset{\mathbf{w} \in \mathcal{W}}{\operatorname{argmin}} \left[ F_S(\mathbf{w}) := \frac{1}{n(n-1)} \sum_{i,j \in [n], i \neq j} f(\mathbf{w}; z_i, z_j) \right]. \tag{1}$$

Our proposed algorithm to solve (1) is described in Algorithm 1. The notation $\Pi_{\mathcal{W}}(\cdot)$ there denotes the projection operator to $\mathcal{W}$. In particular, at iteration $t$, it randomly selects one instance $z_{i_t}$ from the uniform distribution over $[n]$ and pairs it only with the previous instance $z_{i_{t-1}}$, and then do the gradient descent based on $\nabla f(\mathbf{w}_{t-1}; z_{i_t}, z_{i_{t-1}})$. This is in contrast to the classical SGD for pairwise learning in [22, 37, 48] where the present instance $z_{i_t}$ is paired with all previous instances $\{z_{i_1}, z_{i_2}, \ldots, z_{i_{t-1}}\}$. Note $\mathbf{w}_{t-1}$ depends on $z_{i_{t-1}}$ and then $\nabla f(\mathbf{w}_{t-1}; z_{i_t}, z_{i_{t-1}})$ is not an unbiased estimate of $\nabla F_S(\mathbf{w}_{t-1})$. Therefore, the standard analysis of SGD does not apply. We introduce novel techniques to handle this dependency in our analysis (see more details in Section 4).

**Online Learning Setting.** In the online learning setting where the data $\{z_0, z_1, z_2, \ldots\}$ is assumed i.i.d. from an unknown distribution $\rho$ on $\mathcal{Z}$, the number of iterations of an online algorithm is identical to the size of available data. In the same spirit to Algorithm 1, the pseudo code is given in Algorithm 2. Specifically, upon receiving a datum $z_t$ at the current time $t$, we pair it with $z_{t-1}$ which was revealed at the previous time $t - 1$ and then perform gradient descent based on the gradient $\nabla f(\mathbf{w}_{t-1}; z_t, z_{t-1})$. It aims to minimize the population risk which is defined as $F(\mathbf{w}) = \mathbb{E}_{Z, Z'}[f(\mathbf{w}; Z, Z')]$. Here $\mathbb{E}_{Z, Z'}$ denotes the expectation with respect to (w.r.t.) $Z, Z' \sim \rho$.

It is worth pointing out that online learning [18, 31, 32] in general does not require the i.i.d. assumption on the data and study the regret bounds. In this paper, we mainly consider the statistical performance, measured by excess generalization bounds, of the output $\widetilde{\mathbf{w}}_T$ of Algorithm 2 where the streaming data $\{z_0, z_1, z_2, \ldots\}$ is i.i.d. from the population distribution $\rho$.

**Remark 1.** As discussed above, OGD for pairwise learning was proposed and studied in [22, 37] where the current instance $z_t$ is paired with a buffering set $B_t \subseteq \{z_1, \ldots, z_{t-1}\}$. However, the resultant excess generalization bound is in the form of $\mathcal{O}(\frac{1}{\sqrt{s}} + \frac{1}{\sqrt{n}})$ which indicated that the buffer size $s$ needs to be large enough in order to achieve good generalization. Their analysis does not apply to our case since the buffering set $B_t = \{z_{t-1}\}$ with size $s = 1$ for Algorithm 2. As we show soon in Section 4, we can prove that Algorithm 2, which pairs the current instance $z_t$ with the previous instance $z_{t-1}$, still enjoys optimal statistical performance $\mathcal{O}(1/\sqrt{n})$.

**Remark 2.** The work [27] studied the stability and generalization of an SGD-type algorithm for pairwise learning by randomly generating pairs of instances. Specifically, at time $t$, randomly generating a pair $(z_{i_t}, z_{j_t})$ from a given set of training data $S = \{z_i : i \in [n]\}$ and the subsequent update is given by $\mathbf{w}_t = \mathbf{w}_{t-1} - \eta_t \nabla f(\mathbf{w}_{t-1}; z_{i_t}, z_{j_t})$. In contrast, our algorithm, i.e. Algorithm 1, updates the model parameter based on the pair of the current random instance and the random one generated at the previous time $t - 1$, i.e. $(z_{i_t}, z_{i_{t-1}})$. Furthermore, our work here significantly

differs from [27] in the following aspects. Firstly, the algorithm there by randomly selecting pairs of instances does not work in the online learning setting while ours can seamlessly deal with the streaming data as stated in Algorithm 2. Secondly, regarding the technical analysis, our algorithms are more challenging to analyze than the algorithm in [27]. Indeed, $\nabla f(\mathbf{w}_{t-1}; z_{i_t}, z_{j_t})$ is not an unbiased estimate of $\nabla F_S(\mathbf{w}_{t-1})$ due to the independency between $\mathbf{w}_{t-1}$ and $(i_t, j_t)$. Therefore, the optimization error analysis of the algorithm in [27] is the same as the SGD for pointwise learning. As a comparison, $\nabla f(\mathbf{w}_{t-1}; z_{i_t}, z_{i_{t-1}})$ is a biased estimate of $\nabla F_S(\mathbf{w}_{t-1})$ due to the coupling between $\mathbf{w}_{t-1}$ and $i_{t-1}$. We introduce novel techniques to handle this coupling for both the optimization and generalization analyses. Thirdly, we will soon see below that we provide generalization results for nonsmooth, nonconvex losses and also use Algorithm 1 to develop novel differentially private pairwise learning algorithms while [27] focused on the smooth convex losses in the non-private setting.

**Remark 3.** If we let $\xi_t = (i_t, i_{t-1})$ in Algorithm 1, then $\{\xi_t : t \in \mathbb{N}\}$ forms a Markov Chain as $\xi_t$ only depends on $\xi_{t-1}$ but not on $\{\xi_1, \ldots, \xi_{t-2}\}$. Hence, Algorithm 1 can be regarded as a Markov Chain SGD which was studied in [36]. Despite this similarity, our results differ from [36] in two important aspects. Firstly, we are mainly interested in stability and generalization of Algorithm 1 while [36] focused on the convergence analysis of the Markov Chain SGD. One cannot apply the results in [36] to obtain excess generalization bounds for Algorithm 1 in terms of the population risk as we will show soon in the next section. Secondly, directly applying Theorem 1 in [36] only yields a convergence rate of $\mathcal{O}(1/t^{1-q})$ with some $1/2 < q < 1$ in the convex setting. Our proof for the convergence analysis of Algorithm 1 is much simpler and direct which can yield a faster convergence rate $\mathcal{O}(1/\sqrt{t})$ as shown in Section 4.3.

# 4 Generalization Analysis

The aim for the generalization analysis of Algorithm 1 and Algorithm 2 is the same, i.e. to analyze the excess generalization error $F(\mathbf{w}) - F(\mathbf{w}^*)$ of a model $\mathbf{w}$ measuring its relative behavior w.r.t. the best model $\mathbf{w}^* = \arg\min_{\mathbf{w} \in \mathcal{W}} F(\mathbf{w})$.

For Algorithm 1 where the training data $S$ with $n$ datum is given beforehand, the excess generalization involves the generalization error and optimization error. Specifically, one has the following error decomposition for $\bar{\mathbf{w}}_T$

$$\mathbb{E}\big[F(\bar{\mathbf{w}}_T)\big] - F(\mathbf{w}^*) = \mathbb{E}\big[F(\bar{\mathbf{w}}_T) - F_S(\bar{\mathbf{w}}_T)\big] + \mathbb{E}\big[F_S(\bar{\mathbf{w}}_T) - F_S(\mathbf{w}^*)\big]. \tag{2}$$

Here, the expectation is taken w.r.t. the randomness of Algorithm 1, i.e., $\{i_t\}$ and the randomness of data $S$ which is i.i.d. from $\rho$ on $\mathcal{Z}$. We refer to the first term $\mathbb{E}\big[F(\bar{\mathbf{w}}_T) - F_S(\bar{\mathbf{w}}_T)\big]$ as the generalization error and $\mathbb{E}\big[F_S(\bar{\mathbf{w}}_T) - F_S(\mathbf{w}^*)\big]$ as the optimization error. We will use algorithmic stability to handle its generalization error in Subsection 4.2 for smooth and nonsmooth losses. The estimation of the optimization error is given in Subsection 4.3 for both convex and nonconvex losses. For Algorithm 2, there is no generalization error as the data $\{z_1, z_2, \ldots, z_T\}$ is arriving in a sequential manner with $T$ increasing all the time which does not involve the training data. The randomness of Algorithm 2 is only from the i.i.d. data. Therefore, the optimization error in this setting is exactly the excess generalization error $F(\widetilde{\mathbf{w}}_T) - F(\mathbf{w}^*)$ which is estimated in Subsection 4.3.

## 4.1 Excess Generalization Error

In this subsection, we present excess generalization error bounds of Algorithm 1 in terms of the sample size, iteration number and step size, which shows how to tune these parameters to get a model with good generalization. Our analysis requires the following assumptions.

**Assumption.** Let $f : \mathcal{W} \times \mathcal{Z} \times \mathcal{Z} \to \mathbb{R}^+$ and let $\|\cdot\|_2$ denote the Euclidean norm.

(A1) Assume, for any $z, z'$ and $\mathbf{w} \in \mathcal{W}$, that $f(\cdot; z, z')$ is $G$-Lipschitz continuous, i.e. $|f(\mathbf{w}; z, z') - f(\mathbf{w}'; z, z')| \le G\|\mathbf{w} - \mathbf{w}'\|_2$.

(A2) Assume, for any $z, z' \in \mathcal{Z}$, the map $\mathbf{w} \mapsto f(\mathbf{w}; z, z')$ is $L$-strongly smooth, i.e. $f(\mathbf{w}; z, z') - f(\mathbf{w}'; z, z') - \langle \nabla f(\mathbf{w}'; z, z'), \mathbf{w} - \mathbf{w}' \rangle \le \frac{L}{2}\|\mathbf{w} - \mathbf{w}'\|_2^2$.

(A3) Assume, for any $z, z' \in \mathcal{Z}$, $f(\cdot; z, z')$ is $\alpha$-strongly convex, i.e. $f(\mathbf{w}; z, z') - f(\mathbf{w}'; z, z') - \langle \nabla f(\mathbf{w}'; z, z'), \mathbf{w} - \mathbf{w}' \rangle \ge \frac{\alpha}{2}\|\mathbf{w} - \mathbf{w}'\|_2^2$. The case of $\alpha = 0$ is identical to convexity.

(**A4**) Assume $F_S$ satisfies the Polyak-Łojasiewicz (PL) condition with parameter $\mu > 0$, i.e., for $\mathbf{w}_S \in \arg\min_{\mathbf{w} \in \mathcal{W}} F_S(\mathbf{w})$, there holds $2\mu\big(F_S(\mathbf{w}) - F_S(\mathbf{w}_S)\big) \leq \|\nabla F_S(\mathbf{w})\|_2^2$ for all $\mathbf{w} \in \mathcal{W}$.

The PL condition (**A4**) means that the suboptimality in terms of function values can be bounded by gradients [23]. Functions under the PL condition have found various applications including neural networks, matrix factorization, generalized linear models and robust regression (see, e.g., [23]). In particular, AUC maximization problem with the classifier given by a one hidden layer network satisfies the PL condition as shown in [29].

We first study smooth and non-smooth problems for the convex case, and derive the excess generalization bounds of the order $\mathcal{O}(1/\sqrt{n})$ in both cases. We use the notation $B \asymp \tilde{B}$ if there exist constants $c_1, c_2 > 0$ such that $c_1 \tilde{B} \leq B \leq c_2 \tilde{B}$. The proofs are given in Section C.

**Theorem 1** (Nonsmooth Problems). *Let $\mathbf{w}_{-1} = \mathbf{w}_0$ and $\{\mathbf{w}_t : t \in [T]\}$ be produced by Algorithm 1 with $\eta_t = \eta > 0$. Let $\bar{\mathbf{w}}_T = \sum_{t=1}^T \eta_t \mathbf{w}_{t-2} / \sum_{t=1}^T \eta_t$. Let (A1) and (A3) hold true with $\alpha = 0$. Then, we have*

$$\mathbb{E}_{S,\mathcal{A}}[F(\bar{\mathbf{w}}_T)] - F(\mathbf{w}^*) = \mathcal{O}\Big(\sqrt{T}\eta + \frac{T\eta}{n} + \frac{1+T\eta^2}{T\eta}\Big). \tag{3}$$

*Furthermore, selecting $T \asymp n^2$ and $\eta \asymp T^{-\frac{3}{4}}$ yields that $\mathbb{E}_{S,\mathcal{A}}[F(\bar{\mathbf{w}}_T)] - F(\mathbf{w}^*) = \mathcal{O}(1/\sqrt{n})$.*

**Theorem 2** (Smooth Problems). *Let (A1), (A2) and (A3) hold true with $\alpha = 0$. Let $\mathbf{w}_{-1} = \mathbf{w}_0$ and $\{\mathbf{w}_t : t \in [T]\}$ be produced by Algorithm 1 with $\eta_t = \eta \leq 2/L$. Let $\bar{\mathbf{w}}_T = \sum_{t=1}^T \eta_t \mathbf{w}_{t-2} / \sum_{t=1}^T \eta_t$. Then, there holds*

$$\mathbb{E}_{S,\mathcal{A}}[F(\bar{\mathbf{w}}_T)] - F(\mathbf{w}^*) = \mathcal{O}\Big(\frac{T\eta}{n} + \frac{1+T\eta^2}{T\eta}\Big). \tag{4}$$

*Furthermore, choosing $T \asymp n$ and $\eta \asymp T^{-\frac{1}{2}}$ implies that $\mathbb{E}_{S,\mathcal{A}}[F(\bar{\mathbf{w}}_T)] - F(\mathbf{w}^*) = \mathcal{O}(1/\sqrt{n})$.*

**Remark 4.** Notice that the gradient complexity (i.e. the number of computing gradients) of Algorithm 1 is identical to the number of iterations $T$. The above results show, to get excess generalization bounds $\mathcal{O}(1/\sqrt{n})$, that Algorithm 1 requires a gradient complexity $\mathcal{O}(n^2)$ for nonsmooth problems, and $\mathcal{O}(n)$ for smooth problems. This matches the existing generalization analysis for pointwise learning [3, 17, 25]. In Appendix E, additional results are provided where we propose Algorithm 4 based on the iterative localization technique [13] in order to reduce the gradient complexity $\mathcal{O}(n^2)$ required in Theorem 1 to $\mathcal{O}(n)$ for nonsmooth problems.

**Remark 5.** As stated in the introduction, Algorithm 1 can be considered as a specific case of the classic pairwise learning algorithm [22] with a FIFO buffering set $B_{t-1}$ of size $s = 1$. A key difficulty in the generalization analysis is that $\mathbf{w}_{t-1}$ depends on $B_{t-1}$, which renders the standard martingale analysis not applicable. Kar et al. [22] proposed to remove this coupling effect by considering $\sup_{\mathbf{w}} \big[ f(\mathbf{w}; B_{t-1}) - F_S(\mathbf{w}) \big]$, which is why they only derived the excess generalization error bound $\mathcal{O}(1/\sqrt{s})$. We introduce novel techniques to handle the coupling in both generalization analysis and optimization error analysis. For the generalization analysis, our strategy is to write the stability as a deterministic function of several indicator functions on whether we select the different point in neighboring datasets, and then finally consider the randomness of these indicator functions. This delay of considering expectation successfully decouples the coupling between $\mathbf{w}_{t-1}$ and $B_{t-1}$. For the optimization error analysis, our novelty is to observe that $f(\mathbf{w}_{t-1}; z_{i_t}, z_{i_{t-1}}) = f(\mathbf{w}_{t-2}; z_{i_t}, z_{i_{t-1}}) + \mathcal{O}(\eta_{t-1})$, which removes the decoupling since $\mathbf{w}_{t-2}$ is now independent of both $i_t$ and $i_{t-1}$. Since the additional term $\mathcal{O}(\eta_{t-1})$ here is a term of smaller magnitude, the coupling effect is removed without incurring any additional cost.

Finally, we study nonconvex pairwise learning under the PL condition. The proof is in Section C.

**Theorem 3.** *Let (A1), (A2) and (A4) hold true. Let $\alpha_0, B > 0$. Let $\mathbf{w}_{-1} = \mathbf{w}_0$ and $\{\mathbf{w}_j : j \in [t]\}$ be produced by Algorithm 1 with $\eta_j = 2/(\mu(j+1))$. If $\mathbb{E}_{i_{j+1}, i_{j+2}}\big[\|\nabla f(\mathbf{w}_j; z_{i_{j+1}}, z_{i_{j+2}})\|_2^2\big] \leq \alpha_0^2$ and $\sup_{z,z'} f(\mathbf{w}_j; z, z') \leq B$ for any $j$, then*

$$\mathbb{E}\big[F(\mathbf{w}_T)\big] - F(\mathbf{w}^*) = \mathcal{O}\Big(\frac{T^{\frac{2L}{2L+\mu}}}{n}\Big) + \mathcal{O}\big(1/(T\mu^2)\big).$$

*Furthermore, choosing $T \asymp n^{\frac{2L+\mu}{4L+\mu}} \mu^{-\frac{4L+2\mu}{4L+\mu}}$ yields that $\mathbb{E}\big[F(\mathbf{w}_T)\big] - F(\mathbf{w}^*) = n^{-\frac{2L+\mu}{4L+\mu}} \mu^{-\frac{4L}{4L+\mu}}$.*

**Remark 6.** As pointed out before, there is no generalization error for the OGD algorithm, i.e. Algorithm 2 as the i.i.d. data is given in a streaming manner and the iteration number equals the number of the available data (i.e. $t = n$). In this setting, the optimization error is identical to the excess generalization error, which will be estimated in Subsection 4.3.

## 4.2 Stability and Generalization Errors

We study generalization errors by algorithmic stability, which measures the sensitivity of the output of an algorithm w.r.t. the perturbation of the dataset. Below we give the definition of uniform argument stability. We say $S, S'$ are neighboring datasets if they differ at most by a single example.

**Definition 1.** A (randomized) algorithm $\mathcal{A}$ for pairwise learning is called $\varepsilon$-uniformly argument stable if for all neighboring datasets $S, S' \in \mathcal{Z}^n$ we have $\mathbb{E}_{\mathcal{A}}[\|\mathcal{A}(S) - \mathcal{A}(S')\|_2] \leq \varepsilon$.

It is clear $\varepsilon$-uniform argument stability implies $G\varepsilon$-uniform stability i.e., $\sup_{\mathbf{z},\mathbf{z}'} \mathbb{E}_{\mathcal{A}}[f(\mathcal{A}(S), \mathbf{z}, \mathbf{z}') - f(\mathcal{A}(S'), \mathbf{z}, \mathbf{z}')] \leq G\varepsilon$ for Lipschitz losses [7]. The connection between the uniform stability for pairwise learning and its generalization has been established in the literature [1, 34].

**Lemma 1.** If an algorithm $\mathcal{A}$ for pairwise learning is $\varepsilon$-uniformly stable for some $\varepsilon > 0$, then we have $|\mathbb{E}_{S,\mathcal{A}}[F_S(\mathcal{A}(S)) - F(\mathcal{A}(S))]| \leq 2\varepsilon$.

We develop uniform argument stability bound of Algorithm 1 and apply it together with Lemma 1 to establish the following generalization bounds. Theorem 4 handles nonsmooth problems, while Theorem 5 handles smooth problems. The detailed proofs for them can be found in Section A.

**Theorem 4.** Let $\mathbf{w}_{-1} = \mathbf{w}_0$ and $\{\mathbf{w}_j : j \in [t]\}$ be produced by Algorithm 1 with $\eta_j = \eta$. Let (A1) and (A3) hold with $\alpha = 0$. Then, Algorithm 1 is $2\sqrt{e}G\eta\big(\sqrt{5t} + \frac{2t}{n}\big)$-uniformly argument stable and

$$\mathbb{E}_{S,\mathcal{A}}[F(\bar{\mathbf{w}}_t) - F_S(\bar{\mathbf{w}}_t)] \leq 4\sqrt{e}G^2\eta\Big(\sqrt{5t} + \frac{2t}{n}\Big).$$

**Theorem 5.** Let $\mathbf{w}_{-1} = \mathbf{w}_0$ and $\{\mathbf{w}_j : j \in [t]\}$ be produced by Algorithm 1 with $\eta_j = \eta \leq 2/L$. Let (A1), (A2) and (A3) hold true with $\alpha = 0$. Then, Algorithm 1 is $\frac{4G}{n}\sum_{j=1}^{t}\eta_j$-uniformly argument stable and the generalization error satisfies $\mathbb{E}_{S,\mathcal{A}}[F(\bar{\mathbf{w}}_t) - F_S(\bar{\mathbf{w}}_t)] \leq \frac{8G^2}{n}\sum_{j=1}^{t}\eta_j$.

For Algorithm 2, there is no generalization error as the data $\{z_1, z_2, \ldots, z_T\}$ is assumed to arrive in a sequential manner with $T$ increasing all the time which does not involve the training data.

## 4.3 Optimization Error

In this subsection, we establish the convergence rate, i.e., optimization error, of Algorithm 1 for convex, nonconvex and strongly convex problems. We consider both bounds in expectation and with high probability. Our analysis is based on the key observation $f(\mathbf{w}_{t-1}; z_{i_t}, z_{i_{t-1}}) = f(\mathbf{w}_{t-2}; z_{i_t}, z_{i_{t-1}}) + \mathcal{O}(\eta_{t-1})$. The proofs of results in this subsection are given in Section B.

Below we only present optimization error bounds for Algorithm 1 here in the offline (finite-sum) setting where the training data of size $n$, denoted by $S = \{z_1, \ldots, z_n\}$, is fixed, and the optimization error is measured by $F_S(\bar{\mathbf{w}}_t) - \inf_{\mathbf{w} \in \mathcal{W}} F_S(\mathbf{w})$. We emphasize that all our optimization error bounds hold true for Algorithm 2 in the online learning setting with exactly the same analysis where the streaming data $\{z_1, \ldots, z_t, \ldots\}$ is assumed to be i.i.d according to the population distribution $\rho$, and the bounds for the optimization error in this case is given for the excess generalization error (excess population risk), i.e. $F(\widetilde{\mathbf{w}}_t) - \inf_{\mathbf{w} \in \mathcal{W}} F(\mathbf{w})$.

**Theorem 6.** Let $\mathbf{w}_{-1} = \mathbf{w}_0$ and $\{\mathbf{w}_j : j \in [t]\}$ be produced by Algorithm 1. Let (A1) and (A3) hold true with $\alpha \geq 0$. Then, for any $\mathbf{w}$ independent of $\mathcal{A}$ we have the following convergence rates:

(a) Assume $f$ is convex, i.e. $\alpha = 0$. Then, we have

$$\mathbb{E}_{\mathcal{A}}[F_S(\bar{\mathbf{w}}_t)] - F_S(\mathbf{w}) \leq \frac{\|\mathbf{w}_0 - \mathbf{w}\|_2^2 + G^2\sum_{j=1}^{t}(2\eta_j\eta_{j-1} + \eta_j^2)}{2\sum_{j=1}^{t}\eta_j}. \quad (5)$$

(b) Let $f$ be $\alpha$-strongly convex with $\alpha > 0$ and $\eta_j = \frac{2}{\alpha(j+1)}$. Then, there holds $\mathbb{E}_{\mathcal{A}}[F_S(\bar{\mathbf{w}}_t)] - F_S(\mathbf{w}) = \mathcal{O}\big(G^2/(\alpha t)\big)$.

**Remark 7.** The above convergence rates match those in the pointwise learning [5]. Furthermore, if $\eta_j = \eta$, then Eq. (5) becomes $\mathbb{E}_{\mathcal{A}}[F_S(\bar{\mathbf{w}}_t)] - F_S(\mathbf{w}) = \mathcal{O}(1/(t\eta) + \eta)$ and one can choose $\eta \asymp 1/\sqrt{t}$ to get $\mathbb{E}_{\mathcal{A}}[F_S(\bar{\mathbf{w}}_t)] - F_S(\mathbf{w}) = \mathcal{O}(1/\sqrt{t})$. We can extend our convergence analysis to a more general update as $\mathbf{w}_t = \Pi_{\mathcal{W}}\big(\mathbf{w}_{t-1} - \frac{\eta_t}{s}\sum_{j=1}^s \nabla f(\mathbf{w}_{t-1}; z_{i_t}, z_{i_{t-j}})\big)$ for $s \in \mathbb{N}$. Indeed, one can use the observation $f(\mathbf{w}_{t-1}; z_{i_t}, z_{i_{t-s}}) = f(\mathbf{w}_{t-s-1}; z_{i_t}, z_{i_{t-s}}) + \mathcal{O}\big(\sum_{j=1}^s \eta_{t-j}\big)$ to derive the convergence rate $\mathcal{O}(\sqrt{s}/\sqrt{t})$.

Below we present high-probability bounds to understand the variation of the algorithm. We need to take conditional expectation of $f(\mathbf{w}_{2j-2}; z_{i_{2j}}, z_{i_{2j-1}})$ w.r.t. $(i_{2j}, i_{2j-1})$ to get $F_S(\mathbf{w}_{2j-2})$. However, there is a coupling between $(i_{2j}, i_{2j-1})$ and $(i_{2j-1}, i_{2j-2})$. Therefore, one can not directly apply concentration inequalities for martingales to handle $\sum_{j=1}^t \big(f(\mathbf{w}_{2j-2}; z_{i_{2j}}, z_{i_{2j-1}}) - F_S(\mathbf{w}_{2j-2})\big)$. We introduce a novel decoupling technique to handle this coupling. Note that the high-probability bounds match the bounds in expectation up to a constant factor.

**Theorem 7.** *Let $\mathbf{w}_{-1} = \mathbf{w}_0$ and $\{\mathbf{w}_j : j \in [t]\}$ be produced by Algorithm 1. Let (A1) and (A3) hold true with $\alpha \geq 0$ and $\sup_{\mathbf{w}} f(\mathbf{w}; z, z') \leq B$ for some $B > 0$. Let $\delta \in (0, 1)$.*

(a) *Assume $f$ is convex, i.e. $\alpha = 0$ and let $\bar{\mathbf{w}}_t = \sum_{j=1}^t \eta_j \mathbf{w}_{j-2} / \sum_{j=1}^t \eta_j$. Then, for any $\mathbf{w} \in \mathcal{W}$, with probability at least $1 - \delta$ the following inequality holds*

$$F_S(\bar{\mathbf{w}}_t) - F_S(\mathbf{w}) \leq \frac{1}{\sum_{j=1}^t \eta_j}\Big(2B\Big(2\sum_{j=1}^t \eta_j^2 \log(2/\delta)\Big)^{\frac{1}{2}} + \frac{1}{2}\|\mathbf{w}_1 - \mathbf{w}\|_2^2 + G^2 \sum_{j=1}^t \big(\eta_{j-1}\eta_j + \frac{\eta_j^2}{2}\big)\Big).$$

(b) *Assume $f$ is $\alpha$-strongly convex with $\alpha > 0$ and $\eta_j = \frac{2}{\alpha(j+1)}$. Let $\bar{\mathbf{w}}_t = \sum_{j=1}^t j\mathbf{w}_{j-2}/\sum_{j=1}^t j$. Then, with probability at least $1 - \delta$, we have $F_S(\bar{\mathbf{w}}_t) - F_S(\mathbf{w}) = \mathcal{O}\big(G^2\log(1/\delta)/(\alpha t)\big)$.*

Finally, we study the convergence of Algorithm 1 associated with nonconvex functions. We first consider general smooth problems. Since we cannot find a global minimum in this setting, we measure the convergence rate in terms of gradient norms [16]. The following theorem establishes the convergence rate $\mathcal{O}(1/\sqrt{t})$ for $\min_{j=1,\ldots,t} \mathbb{E}_{\mathcal{A}}[\|\nabla F_S(\mathbf{w}_j)\|_2^2]$.

**Theorem 8.** *Let $\mathbf{w}_{-1} = \mathbf{w}_0$ and $\{\mathbf{w}_j : j \in [t]\}$ be produced by Algorithm 1 with $\eta_j = \eta \leq 1/(2\sqrt{L})$. Let (A2) hold true and $\mathbb{E}_{i_{j+1}, i_{j+2}}\big[\|\nabla f(\mathbf{w}_j; z_{i_{j+1}}, z_{i_{j+2}})\|_2^2\big] \leq \alpha_0^2$ for some $\alpha_0$ and any $j$. Then,*

$$\frac{1}{t}\sum_{j=1}^t \mathbb{E}_{\mathcal{A}}\big[\|\nabla F_S(\mathbf{w}_{j-2})\|_2^2\big] \leq \frac{F_S(\mathbf{w}_0)}{t\eta} + 8L\eta\alpha_0^2.$$

*Furthermore, choosing $\eta \asymp 1/\sqrt{t}$ implies that $\frac{1}{t}\sum_{j=1}^t \mathbb{E}_{\mathcal{A}}\big[\|\nabla F_S(\mathbf{w}_{j-2})\|_2^2\big] = \mathcal{O}(1/\sqrt{t})$.*

We now turn to nonconvex problems under a PL condition. Theorem 9 gives convergence rates of the order $\mathcal{O}(1/t)$, which match the existing results for standard SGD in pointwise learning [23].

**Theorem 9.** *Assume (A2) and (A4) hold true and $\mathbb{E}_{i_{j+1}, i_{j+2}}\big[\|\nabla f(\mathbf{w}_j; z_{i_{j+1}}, z_{i_{j+2}})\|_2^2\big] \leq \alpha_0^2$ for some $\alpha_0$ and any $j$. Let $\mathbf{w}_{-1} = \mathbf{w}_0$ and $\{\mathbf{w}_j : j \in [t]\}$ be produced by Algorithm 1 with $\eta_j = 2/(\mu(j+1))$. Then*

$$\mathbb{E}_{\mathcal{A}}[F_S(\mathbf{w}_t) - F_S(\mathbf{w}_S)] \leq \frac{32L\alpha_0^2}{\mu^2}\Big(\frac{1}{t+1} + \frac{\log(et)}{\mu t(t+1)}\Big).$$

# 5 Application: Differentially Private SGD for Pairwise Learning

We now use Algorithm 1 and our stability analysis (i.e. Theorem 5) to develop a differentially private algorithm for pairwise learning. Let us start with the definition of differential privacy [12].

**Definition 2.** *A (randomized) algorithm $\mathcal{A}$ is called $(\epsilon, \delta)$-differentially private (DP) if, for all neighboring datasets $S, S'$ and for all events $O$ in the output space of $\mathcal{A}$, one has $\mathbb{P}[\mathcal{A}(S) \in O] \leq e^{\epsilon}\mathbb{P}[\mathcal{A}(S') \in O] + \delta$.*

---

**Algorithm 3** Differentially Private Localized SGD for Pairwise Learning

---

1: **Inputs:** Dataset $S = \{\mathbf{z}_i : i \in [n]\}$, parameters $\epsilon, \delta > 0$, and learning rate $\eta$, initial point $\mathbf{w}_0$
2: Set $K = \lceil \log_2 n \rceil$ and divide $S$ into $K$ disjoint subsets $\{S_1, \cdots, S_K\}$ where $|S_k| = n_k = 2^{-k}n$.
3: **for** $k = 1$ to $K$ **do**
4:      Set $\eta_k = 4^{-k}\eta$
5:      Compute $\bar{\mathbf{w}}_k$ by Algorithm 1 based on $S_k$ and initiated at $\mathbf{w}_{k-1}$ for $\lceil n_k \log(4/\delta) \rceil$ steps.
6:      Set $\mathbf{w}_k = \bar{\mathbf{w}}_k + \mathbf{u}_k$ where $\mathbf{u}_k \sim \mathcal{N}(0, \sigma_k^2 I_d)$ with $\sigma_k = 12G\eta_k \log(4/\delta)\sqrt{2\log(2.5/\delta)}/\epsilon$.
7: **Outputs:** $\mathbf{w}_K$

---

Our proposed DP algorithm for pairwise learning is described in Algorithm 3 which is inspired by the iterative localization technique [13] for pointwise learning. The privacy and utility guarantees are given by the following theorem. Here $D$ denotes the diameter of $\mathcal{W}$.

**Theorem 10.** *Let (A1), (A2), and (A3) hold true with $\sigma = 0$. Let $\{\mathbf{w}_k : k \in [K]\}$ be produced by Algorithm 3 with $\eta = \frac{D}{G} \min\{\frac{\log(4/\delta)}{\sqrt{n}}, \frac{\epsilon}{12\log(4/\delta)\sqrt{2d\log(2.5/\delta)}}\} \leq \frac{2}{L}$. Then, Algorithm 3 satisfies $(\epsilon, \delta)$-DP and, with gradient complexity $\mathcal{O}(n\log(1/\delta))$, we have the utility bound that*

$$\mathbb{E}[F(\mathbf{w}_K) - F(\mathbf{w}^*)] = \mathcal{O}\Big(GD\Big(\frac{1}{\sqrt{n}} + \frac{\sqrt{d}\log^{\frac{3}{2}}(1/\delta)}{\epsilon n}\Big)\Big).$$

The main difference from the pointwise setting in [13] is that Algorithm 3 involves the coupling dependency between $\{i_t, i_{t-1}\}$ at time $t$ in Algorithm 1 and $\{i_{t-1}, i_{t-2}\}$ at time $t-1$, which renders the direct application of the standard concentration inequalities infeasible. We propose a novel decomposition to circumvent this hurdle (see more detailed proof for Theorem 10 in Appendix D).

**Remark 8.** The above bound matches the lower bound given in [2] for $(\epsilon, \delta)$-differentially private pointwise learning up to a $\log(1/\delta)$ term. Our utility bound improves over the previous work [21] which has the bound $\mathcal{O}(\sqrt{d\log(1/\delta)}\log(n/\delta)/(\sqrt{n}\epsilon))$. During the preparation of this work, we notice a very recent paper [42] also studied the private version of pairwise algorithm by using the localization technique. Their algorithm establishes the optimal rate $\mathcal{O}(1/\sqrt{n} + \sqrt{d\log(1/\delta)}/(\epsilon n))$ which, however, needs an expensive gradient complexity $\mathcal{O}(n^3\log(1/\delta))$. As a comparison, we achieve nearly optimal utility bound with linear gradient complexity $\mathcal{O}(n\log(1/\delta))$.

**Remark 9.** In Appendix F, we further remove the smoothness assumption (A2) required in Theorem 10 and propose a private algorithm (stated as Algorithm 5 there) that achieves the optimal rate $\mathcal{O}\big((1/\sqrt{n} + \sqrt{d\log(1/\delta)}/(\epsilon n))\big)$ with gradient complexity $\mathcal{O}(n^2\log(1/\delta))$. Such bound improves over the previous known results with nonsmooth losses [43] where the utility bound was $\mathcal{O}(\sqrt{d\log(1/\delta)}\log(n/\delta)/(\sqrt{n}\epsilon))$.

## 6 Experimental Validation

We now report some preliminary experiments[2] on AUC maximization with $f(\mathbf{w}; (\mathbf{x}, y), (\mathbf{x}', y')) = \ell(\mathbf{w}^\top(\mathbf{x} - \mathbf{x}'))\mathbb{I}_{[y=1 \wedge y'=-1]}$ where $\ell$ is a surrogate loss function, e.g., the hinge loss $\ell(t) = (1-t)_+$.

The purpose of our first experiment is to compare our algorithm, i.e. Algorithm 1, against four existing algorithms for pairwise learning in terms of generalization and CPU running time on several datasets available from the LIBSVM website [9]. These algorithms are: 1) OLP [22] uses a buffer $B_t$ updated by a variant of Reservoir sampling with replacement where the buffer size is chosen to be 200 in order to guarantee the maximum AUC score as indicated in [22]; 2) OAM$_{gra}$ [48] is tailored for AUC maximization with the hinge loss which uses buffers by Reservoir sampling. The buffer size is set to be 100 for both positive and negative buffers as suggested in that paper; 3) SGD$_{pair}$ [27] randomly pick a pair from $\binom{n}{2}$ pairs by uniform distribution; 4) SPAUC [26], where AUC maximization problem with the least square loss was reformulated as stochastic saddle point (min-max) problem. Note that SPAUC and OAM$_{gra}$ can only apply to AUC maximization problem with the least square loss and hinge loss, respectively.

---

[2]The source codes are available at https://github.com/zhenhuan-yang/simple-pairwise.

Table 1: Average AUC score $\pm$ standard deviation across multiple datasets. Our best results are highlighted in bold.

| Algorithm | diabetes | german | ijcnn1 | letter | mnist | usps |
|---|---|---|---|---|---|---|
| Ours | $\mathbf{.831 \pm .030}$ | $.793 \pm .021$ | $\mathbf{.934 \pm .002}$ | $.810 \pm .007$ | $\mathbf{.932 \pm .001}$ | $\mathbf{.926 \pm .006}$ |
| $\text{SGD}_{pair}$ [27] | $.830 \pm .028$ | $.794 \pm .023$ | $.934 \pm .003$ | $.811 \pm .008$ | $.932 \pm .001$ | $.925 \pm .006$ |
| OLP [22] | $.825 \pm .028$ | $.787 \pm .028$ | $.916 \pm .003$ | $.808 \pm .010$ | $.927 \pm .003$ | $.917 \pm .006$ |
| $\text{OAM}_{gra}$ [48] | $.828 \pm .026$ | $.785 \pm .029$ | $.930 \pm .003$ | $.806 \pm .008$ | $.898 \pm .002$ | $.916 \pm .005$ |
| SPAUC [26] | $.828 \pm .031$ | $.799 \pm .026$ | $.932 \pm .002$ | $.809 \pm .008$ | $.927 \pm .002$ | $.923 \pm .005$ |

To validate the generalization ability, the surrogate loss for Algorithm 1, $\text{SGD}_{pair}$ and OLP is chosen to be the hinge loss. Average AUC scores of different algorithms are listed in Table 1 where we can see that our algorithm yields competitive generalization performance with OAM and OLP using a large buffering set. Detailed experimental setup, data statistics and more results such as comparison with OLP and OAM with the size of the buffering set $s = 1$ are listed in Appendix G.

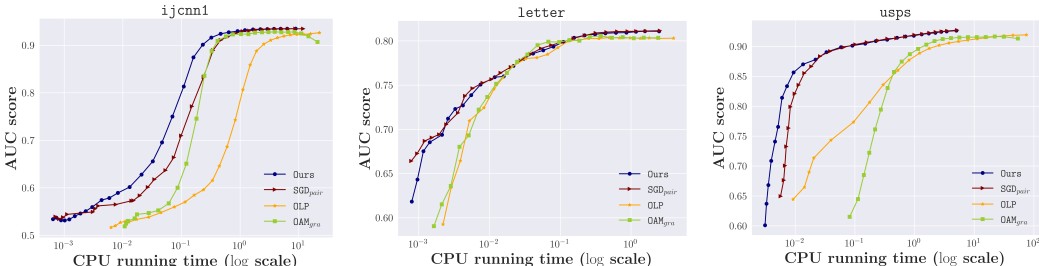

Figure 1: CPU running time ($\log$ scale) versus the AUC score

To fairly compare the CPU running time, we apply the following uniform setting across all algorithms: 1) $\mathcal{W}$ is an $\ell_2$ ball with the same diameter; 2) the step sizes $\eta_t = \eta$ which is tuned by cross validation. We report the results in Figure 1 for the hinge loss. We can see that CPU running time for our algorithm and $\text{SGD}_{pair}$ are similar while OLP and OAM needs more time to converge. The possible reason behind this is that they have a high gradient complexity $\mathcal{O}(s)$ at each iteration while ours is $\mathcal{O}(1)$. More results on comparison for the least square loss and for differentially private algorithms are given in Appendix G.

## 7  Conclusion

In this paper, we propose simple stochastic and online gradient descent algorithms for pairwise learning. The key idea is to build a gradient estimator by pairing the current instance with the previous instance, which enjoys favorable computation and storage complexity. We leverage the lens of algorithmic stability to study its generalization and apply tools in optimization theory to study its convergence rates for various problems including convex/nonconvex and smooth/nonsmooth settings. We also use our algorithms and stability analysis to develop a new DP algorithm for pairwise learning with differential privacy constraints which significantly improves the existing results. The main difference from pointwise learning in the analysis is the coupling between models and previous instances, which is handled by introducing novel decoupling techniques.

For future work, it would be interesting to see whether the analysis and results still hold true if the current example $\mathbf{z}_{i_t}$ in Algorithm 1 is paired with one arbitrary previous example (e.g., $\mathbf{z}_{i_1}$ ). Other future work would be a systematic extension of our algorithms using other acceleration schemes such as momentum and variance reduction techniques.

## Acknowledgments

The authors are grateful to the anonymous reviewers for their constructive comments and suggestions. Tianbao Yang is partially supported by NSF Career Award #1844403 and NSF Award #2110545. Yiming Ying is supported by NSF under grants DMS-2110836, IIS-1816227, IIS-2110546, and IIS-2103450.

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
