# A Proofs of Generalization Error (Theorems 4 and 5)

In this section, we present the stability and generalization error bounds for SGD with convex loss functions. We first prove Theorem 4 on smooth problems and then Theorem 5 on nonsmooth problems. Recall $\mathbb{E}_{\mathcal{A}}$ denotes the expectation w.r.t. the internal randomness of $\mathcal{A}$. For SGD, this means the expectation w.r.t. $\{i_j\}_{j \in [t]}$.

**Proof of Theorem 4.** We first investigate the uniform stability of Algorithm 1. Let $S' = \{z_1, \ldots, z_{n-1}, z'_n\}$, where $z'_n$ is independently drawn from $\rho$, and $\{\mathbf{w}'_t\}$ be produced by Algorithm 1 w.r.t. data $S'$. We consider two cases: i.e. the case of $\{i_t \neq n \text{ and } i_{t-1} \neq n\}$ and the case of $\{i_t = n \text{ or } i_{t-1} = n\}$.

If $i_t \neq n$ and $i_{t-1} \neq n$, then

$$
\begin{aligned}
\left\|\mathbf{w}_t - \mathbf{w}'_t\right\|_2^2 &\leq \left\|\mathbf{w}_{t-1} - \eta_t \nabla f(\mathbf{w}_{t-1}; z_{i_t}, z_{i_{t-1}}) - \mathbf{w}'_{t-1} + \eta_t \nabla f(\mathbf{w}'_{t-1}; z'_{i_t}, z'_{i_{t-1}})\right\|_2^2 \\
&= \left\|\mathbf{w}_{t-1} - \eta_t \nabla f(\mathbf{w}_{t-1}; z_{i_t}, z_{i_{t-1}}) - \mathbf{w}'_{t-1} + \eta_t \nabla f(\mathbf{w}'_{t-1}; z_{i_t}, z_{i_{t-1}})\right\|_2^2 \\
&= \left\|\mathbf{w}_{t-1} - \mathbf{w}'_{t-1}\right\|_2^2 + \eta_t^2 \left\|\nabla f(\mathbf{w}_{t-1}; z_{i_t}, z_{i_{t-1}}) - \nabla f(\mathbf{w}'_{t-1}; z_{i_t}, z_{i_{t-1}})\right\|_2^2 \\
&\quad - 2\eta_t \langle \mathbf{w}_{t-1} - \mathbf{w}'_{t-1}, \nabla f(\mathbf{w}_{t-1}; z_{i_t}, z_{i_{t-1}}) - \nabla f(\mathbf{w}'_{t-1}; z_{i_t}, z_{i_{t-1}}) \rangle \\
&\leq \left\|\mathbf{w}_{t-1} - \mathbf{w}'_{t-1}\right\|_2^2 + \eta_t^2 \left\|\nabla f(\mathbf{w}_{t-1}; z_{i_t}, z_{i_{t-1}}) - \nabla f(\mathbf{w}'_{t-1}; z_{i_t}, z_{i_{t-1}})\right\|_2^2 \\
&\leq \left\|\mathbf{w}_{t-1} - \mathbf{w}'_{t-1}\right\|_2^2 + 4\eta_t^2 G^2,
\end{aligned}
$$

where the last second inequality follows from the inequality $\langle \mathbf{w}_{t-1} - \mathbf{w}'_{t-1}, \nabla f(\mathbf{w}_{t-1}; z_{i_t}, z_{i_{t-1}}) - \nabla f(\mathbf{w}'_{t-1}; z_{i_t}, z_{i_{t-1}}) \rangle \geq 0$ due to the convexity of $f$ and the last inequality follows from the Lipschitz continuity of $f$. If $i_t = n$ or $i_{t-1} = n$, it follows from the elementary inequality $(a+b)^2 \leq (1+p)a^2 + (1+1/p)b^2$ and the Lipschitz condition that

$$
\begin{aligned}
\left\|\mathbf{w}_t - \mathbf{w}'_t\right\|_2^2 &\leq (1+p)\left\|\mathbf{w}_{t-1} - \mathbf{w}'_{t-1}\right\|_2^2 \\
&\quad + (1+1/p)\eta_t^2 \left\|\nabla f(\mathbf{w}_{t-1}; z_{i_t}, z_{i_{t-1}}) - \nabla f(\mathbf{w}'_{t-1}; z'_{i_t}, z'_{i_{t-1}})\right\|_2^2 \\
&\leq (1+p)\left\|\mathbf{w}_{t-1} - \mathbf{w}'_{t-1}\right\|_2^2 + 4(1+1/p)\eta_t^2 G^2.
\end{aligned}
$$

We can combine the above two cases together and derive

$$
\begin{aligned}
\left\|\mathbf{w}_t - \mathbf{w}'_t\right\|_2^2 &\leq \left(\left\|\mathbf{w}_{t-1} - \mathbf{w}'_{t-1}\right\|_2^2 + 4\eta_t^2 G^2\right)\mathbb{I}_{[i_t \neq n \text{ and } i_{t-1} = n]} \\
&\quad + \left((1+p)\left\|\mathbf{w}_{t-1} - \mathbf{w}'_{t-1}\right\|_2^2 + 4(1+1/p)\eta_t^2 G^2\right)\mathbb{I}_{[i_t = n \text{ or } i_{t-1} = n]} \\
&\leq \left(1 + p\mathbb{I}_{[i_t = n \text{ or } i_{t-1} = n]}\right)\left\|\mathbf{w}_{t-1} - \mathbf{w}'_{t-1}\right\|_2^2 + 4\eta_t^2 G^2\left(1 + \mathbb{I}_{[i_t = n \text{ or } i_{t-1} = n]}/p\right) \\
&= \left(1 + p\right)^{\mathbb{I}_{[i_t = n \text{ or } i_{t-1} = n]}}\left\|\mathbf{w}_{t-1} - \mathbf{w}'_{t-1}\right\|_2^2 + 4\eta_t^2 G^2\left(1 + \mathbb{I}_{[i_t = n \text{ or } i_{t-1} = n]}/p\right),
\end{aligned}
$$

where $\mathbb{I}_{[\cdot]}$ is the indicator function. We can apply the above inequality recursively and get

$$
\begin{aligned}
\left\|\mathbf{w}_t - \mathbf{w}'_t\right\|_2^2 &\leq 4G^2 \sum_{k=1}^{t} \eta_k^2 \left(1 + \mathbb{I}_{[i_k = n \text{ or } i_{k-1} = n]}/p\right) \prod_{j=k+1}^{t} \left(1+p\right)^{\mathbb{I}_{[i_j = n \text{ or } i_{j-1} = n]}} \\
&\leq 4G^2 \prod_{j=1}^{t} \left(1+p\right)^{\mathbb{I}_{[i_j = n \text{ or } i_{j-1} = n]}} \sum_{k=1}^{t} \eta_k^2 \left(1 + \mathbb{I}_{[i_k = n \text{ or } i_{k-1} = n]}/p\right) \\
&= 4G^2 \eta^2 \left(1+p\right)^{\sum_{j=1}^{t} \mathbb{I}_{[i_j = n \text{ or } i_{j-1} = n]}} \left(t + \sum_{k=1}^{t} \mathbb{I}_{[i_k = n \text{ or } i_{k-1} = n]}/p\right),
\end{aligned}
$$

where the last inequality follows from $\eta_j = \eta$.

Now, we choose $p = 1/\left(\sum_{j=1}^{t} \mathbb{I}_{[i_j = n \text{ or } i_{j-1} = n]}\right)$ and use the inequality $(1+x)^{1/x} \leq e$ to derive the following inequality

$$
\left\|\mathbf{w}_t - \mathbf{w}'_t\right\|_2^2 \leq 4eG^2 \eta^2 \left(t + \left(\sum_{k=1}^{t} \mathbb{I}_{[i_k = n \text{ or } i_{k-1} = n]}\right)^2\right).
$$

By the inequality $\mathbb{I}_{[i_k=n \text{ or } i_{k-1}=n]} \leq \mathbb{I}_{[i_k=n]} + \mathbb{I}_{[i_{k-1}=n]}$ and $(a+b)^2 \leq 2(a^2 + b^2)$ we know

$$\mathbb{E}_{\mathcal{A}}\Big[\Big(\sum_{k=1}^{t}\mathbb{I}_{[i_k=n \text{ or } i_{k-1}=n]}\Big)^2\Big] \leq 2\mathbb{E}_{\mathcal{A}}\Big[\Big(\sum_{k=1}^{t}\mathbb{I}_{[i_k=n]}\Big)^2\Big] + 2\mathbb{E}_{\mathcal{A}}\Big[\Big(\sum_{k=1}^{t}\mathbb{I}_{[i_{k-1}=n]}\Big)^2\Big]$$

$$= 4\mathbb{E}\Big[\Big(\sum_{k=1}^{t}\mathbb{I}_{[i_k=n]}\Big)^2\Big] \leq 4t + 4\sum_{j,k\in[t]:j\neq k}\mathbb{E}\big[\mathbb{I}_{[i_j=n]}\mathbb{I}_{[i_k=n]}\big]$$

$$= 4t + 4\sum_{j,k\in[t]:j\neq k}\frac{1}{n^2} \leq 4t + 4t^2/n^2.$$

We can combine the above two inequalities together and derive

$$\mathbb{E}_{\mathcal{A}}[\|\mathbf{w}_t - \mathbf{w}_t'\|_2^2] \leq 4eG^2\eta^2\Big(5t + \frac{4t^2}{n^2}\Big)$$

and by the convexity of $\|\cdot\|_2^2$ it follows

$$\mathbb{E}_{\mathcal{A}}[\|\bar{\mathbf{w}}_t - \bar{\mathbf{w}}_t'\|_2^2] \leq \frac{1}{t}\sum_{j=1}^{t}\mathbb{E}_{\mathcal{A}}[\|\mathbf{w}_j - \mathbf{w}_j'\|_2^2] \leq 4eG^2\eta^2\Big(5t + \frac{4t^2}{n^2}\Big).$$

This establishes the uniform stability of Algorithm 1. Furthermore, for any $\mathbf{z}, \mathbf{z}'$, we have

$$\mathbb{E}_{\mathcal{A}}[f(\bar{\mathbf{w}}_t, \mathbf{z}, \mathbf{z}') - f(\bar{\mathbf{w}}_t', \mathbf{z}, \mathbf{z}')] \leq G\mathbb{E}_{\mathcal{A}}[\|\bar{\mathbf{w}}_t - \bar{\mathbf{w}}_t'\|_2] = G\mathbb{E}_{\mathcal{A}}\Big[\sqrt{\|\bar{\mathbf{w}}_t - \bar{\mathbf{w}}_t'\|_2^2}\Big]$$

$$\leq G\sqrt{\mathbb{E}_{\mathcal{A}}[\|\bar{\mathbf{w}}_t - \bar{\mathbf{w}}_t'\|_2^2]} \leq 2\sqrt{e}G^2\eta\Big(\sqrt{5t} + \frac{2t}{n}\Big)$$

where the first inequality we used the $G$-Lipschitz continuity of $f$ and the second inequality we used the Jensen's inequality. Therefore, Algorithm 1 is $2\sqrt{e}G^2\eta\Big(\sqrt{5t} + \frac{2t}{n}\Big)$-uniformly stable. By Lemma 1 it follows

$$\mathbb{E}_{\mathcal{A}}[F(\bar{\mathbf{w}}_t) - F_S(\bar{\mathbf{w}}_t)] \leq 4\sqrt{e}G^2\eta\Big(\sqrt{5t} + \frac{2t}{n}\Big),$$

which gives us the desired result. $\qquad\square$

To prove Theorem 5 we require the following lemma on the nonexpansiveness of gradient map $\mathbf{w} \mapsto \mathbf{w} - \eta\nabla f(\mathbf{w}; z, z')$.

**Lemma 2** (Hardt et al. 17). *Assume for all $z \in \mathcal{Z}$, the function $\mathbf{w} \mapsto f(\mathbf{w}; z, z')$ is convex and $L$-smooth. Then for all $\eta \leq 2/L$ and $z, z' \in \mathcal{Z}$ there holds*

$$\|\mathbf{w} - \eta\nabla f(\mathbf{w}; z, z') - \mathbf{w}' + \eta\nabla f(\mathbf{w}'; z, z')\|_2 \leq \|\mathbf{w} - \mathbf{w}'\|_2.$$

**Proof of Theorem 5.** Let $S' = \{z_1, \ldots, z_{n-1}, z_n'\}$, where $z_n'$ is independently drawn from $\rho$. Let $\{\mathbf{w}_t'\}$ be produced by Algorithm 1 w.r.t. $S'$. We consider two cases. If $i_t \neq n$ and $i_{t-1} \neq n$, then it follows from Lemma 2 that

$$\|\mathbf{w}_t - \mathbf{w}_t'\|_2 \leq \|\mathbf{w}_{t-1} - \eta_t\nabla f(\mathbf{w}_{t-1}; z_{i_t}, z_{i_{t-1}}) - \mathbf{w}_{t-1}' + \eta_t\nabla f(\mathbf{w}_{t-1}'; z_{i_t}', z_{i_{t-1}}')\|_2$$

$$= \|\mathbf{w}_{t-1} - \eta_t\nabla f(\mathbf{w}_{t-1}; z_{i_t}, z_{i_{t-1}}) - \mathbf{w}_{t-1}' + \eta_t\nabla f(\mathbf{w}_{t-1}'; z_{i_t}, z_{i_{t-1}})\|_2$$

$$\leq \|\mathbf{w}_{t-1} - \mathbf{w}_{t-1}'\|_2.$$

Otherwise, we know

$$\|\mathbf{w}_t - \mathbf{w}_t'\|_2 \leq \|\mathbf{w}_{t-1} - \mathbf{w}_{t-1}'\|_2 + \eta_t\|\nabla f(\mathbf{w}_{t-1}; z_{i_t}, z_{i_{t-1}}) - \nabla f(\mathbf{w}_{t-1}'; z_{i_t}', z_{i_{t-1}}')\|_2$$

$$\leq \|\mathbf{w}_{t-1} - \mathbf{w}_{t-1}'\|_2 + 2\eta_t G.$$

We can combine the above two cases together and derive the following inequality

$$\|\mathbf{w}_t - \mathbf{w}_t'\|_2 \leq \|\mathbf{w}_{t-1} - \mathbf{w}_{t-1}'\|_2\mathbb{I}_{[i_t\neq n \text{ and } i_{t-1}\neq n]} + \big(\|\mathbf{w}_{t-1} - \mathbf{w}_{t-1}'\|_2 + 2\eta_t G\big)\mathbb{I}_{[i_t=n \text{ or } i_{t-1}=n]}$$

$$= \|\mathbf{w}_{t-1} - \mathbf{w}_{t-1}'\|_2 + 2\eta_t G\mathbb{I}_{[i_t=n \text{ or } i_{t-1}=n]}.$$

We can apply the above inequality recursively and get

$$\left\|\mathbf{w}_t - \mathbf{w}_t'\right\|_2 \leq 2G \sum_{j=1}^{t} \eta_j \mathbb{I}_{[i_j = n \text{ or } i_{j-1} = n]} \leq 2G \sum_{j=1}^{t} \eta_j \left( \mathbb{I}_{[i_j = n]} + \mathbb{I}_{[i_{j-1} = n]} \right).$$

Taking expectations over both sides gives $\mathbb{E}_{\mathcal{A}}\left[\left\|\mathbf{w}_t - \mathbf{w}_t'\right\|_2\right] \leq \frac{4G}{n} \sum_{j=1}^{t} \eta_j$. It then follows from the convexity of $\|\cdot\|_2$ that

$$\mathbb{E}_{\mathcal{A}}\left[\left\|\bar{\mathbf{w}}_t - \bar{\mathbf{w}}_t'\right\|_2\right] \leq \frac{4G}{n} \sum_{j=1}^{t} \eta_j.$$

This establishes the uniform argument stability of Algorithm 1. Furthermore, it follows the Lipschitz condition that

$$\sup_{z,z'} \mathbb{E}_{\mathcal{A}}\left[f(\bar{\mathbf{w}}_t; z, z') - f(\bar{\mathbf{w}}_t'; z, z')\right] \leq \frac{4G^2}{n} \sum_{j=1}^{t} \eta_j.$$

The desired result then follows from Lemma 1. The proof for Theorem 5 is completed. □

Finally, we consider the generalization analysis for nonconvex problems under the PL condition. To prove Theorem 3, we first introduce a lemma motivated by the arguments in [17].

**Lemma 3.** *Let* $S = \{z_i\}_{i \in [n]}$ *and* $S' = \{z_i'\}_{i \in [n]}$ *be neighboring datasets differing by a single example. Let* $\{\mathbf{w}_t\}_t$ *and* $\{\mathbf{w}_t'\}_t$ *be produced by Algorithm 1 w.r.t.* $S$ *and* $S'$, *respectively. Let Assumption (A1) hold and* $\sup_{z,z'} f(\mathbf{w}_i, z, z') \leq B$. *Let* $\triangle_t = \|\mathbf{w}_t - \mathbf{w}_t'\|_2$. *Then for every* $z, z' \in \mathcal{Z}$ *and every* $t_0 \in [n]$, *there holds*

$$\mathbb{E}\left[|f(\mathbf{w}_T; z, z') - f(\mathbf{w}_T'; z, z')|\right] \leq G\mathbb{E}\left[\triangle_T | \triangle_{t_0} = 0\right] + \frac{Bt_0}{n}.$$

*Proof.* Without loss of generality, we assume that $S$ and $S'$ differ by the last example. Let $\mathcal{E}$ denote the event $\triangle_{t_0} = 0$. Then we have

$$\mathbb{E}\left[|f(\mathbf{w}_T; z, z') - f(\mathbf{w}_T'; z, z')|\right] = \mathbb{E}\left[|f(\mathbf{w}_T; z, z') - f(\mathbf{w}_T'; z, z')||\mathcal{E}\right]\Pr\{\mathcal{E}\}$$
$$+ \mathbb{E}\left[|f(\mathbf{w}_T; z, z') - f(\mathbf{w}_T'; z, z')||\mathcal{E}^c\right]\Pr\{\mathcal{E}^c\},$$

where $\mathcal{E}^c$ denotes the complement of $\mathcal{E}$. Furthermore, we know

$$\Pr\{\mathcal{E}^c\} \leq \sum_{t=1}^{t_0} \Pr\{i_t = n\} = \frac{t_0}{n}.$$

We can combine the above two inequalities and the Lipschitz continuity of $f$ to derive the stated bound, which completes the proof. □

# B  Proofs of Optimization Error (Theorems 6-9)

In this section, we prove optimization error bounds for SGD. We first consider convex cases, and prove convergence rates in expectation (Theorem 6) and with high probability (Theorem 7). Then, we establish convergence rates for SGD with nonconvex loss functions (Theorem 8 and Theorem 9).

**Proof of Theorem 6.** Consider $j \geq 1$. Note that $f(\cdot; z, z')$ is $\alpha$-strongly convex and $G$-Lipschitz continuous, we have

$$
\begin{aligned}
\|\mathbf{w}_j - \mathbf{w}\|_2^2 &\leq \|\mathbf{w}_{j-1} - \eta_j \nabla f(\mathbf{w}_{j-1}; z_{i_j}, z_{i_{j-1}}) - \mathbf{w}\|_2^2 \\
&= \|\mathbf{w}_{j-1} - \mathbf{w}\|_2^2 - 2\eta_j \langle \nabla f(\mathbf{w}_{j-1}; z_{i_j}, z_{i_{j-1}}), \mathbf{w}_{j-1} - \mathbf{w} \rangle + \eta_j^2 \|\nabla f(\mathbf{w}_{j-1}; z_{i_j}, z_{i_{j-1}})\|_2^2 \\
&\leq (1 - \eta_j \alpha)\|\mathbf{w}_{j-1} - \mathbf{w}\|_2^2 - 2\eta_j [f(\mathbf{w}_{j-1}; z_{i_j}, z_{i_{j-1}}) - f(\mathbf{w}; z_{i_j}, z_{i_{j-1}})] + G^2 \eta_j^2 \\
&= (1 - \eta_j \alpha)\|\mathbf{w}_{j-1} - \mathbf{w}\|_2^2 - 2\eta_j [f(\mathbf{w}_{j-2}; z_{i_j}, z_{i_{j-1}}) - f(\mathbf{w}; z_{i_j}, z_{i_{j-1}})] \\
&\quad + 2\eta_j [f(\mathbf{w}_{j-2}; z_{i_j}, z_{i_{j-1}}) - f(\mathbf{w}_{j-1}; z_{i_j}, z_{i_{j-1}})] + G^2 \eta_j^2 \\
&\leq (1 - \eta_j \alpha)\|\mathbf{w}_{j-1} - \mathbf{w}\|_2^2 - 2\eta_j [f(\mathbf{w}_{j-2}; z_{i_j}, z_{i_{j-1}}) - f(\mathbf{w}; z_{i_j}, z_{i_{j-1}})] \\
&\quad + 2\eta_j G \|\mathbf{w}_{j-1} - \mathbf{w}_{j-2}\|_2 + G^2 \eta_j^2 \\
&\leq (1 - \eta_j \alpha)\|\mathbf{w}_{j-1} - \mathbf{w}\|_2^2 - 2\eta_j [f(\mathbf{w}_{j-2}; z_{i_j}, z_{i_{j-1}}) - f(\mathbf{w}; z_{i_j}, z_{i_{j-1}})] \\
&\quad + 2G^2 \eta_j \eta_{j-1} + G^2 \eta_j^2,
\end{aligned}
\tag{B.1}
$$

where the last inequality used the fact that $\|\mathbf{w}_j - \mathbf{w}_{j-1}\|_2 = \eta_j \|\nabla f(\mathbf{w}_j; z_{i_j}, z_{i_{j-1}})\|_2 \leq G\eta_j$.

For the convex case, i.e. $\alpha = 0$, we know from (B.1) that

$$
\begin{aligned}
\sum_{j=1}^{t} &\eta_j [f(\mathbf{w}_{j-2}; z_{i_j}, z_{i_{j-1}}) - f(\mathbf{w}; z_{i_j}, z_{i_{j-1}})] \\
&\leq \frac{1}{2} \sum_{j=1}^{t} [\|\mathbf{w}_{j-1} - \mathbf{w}\|_2^2 - \|\mathbf{w}_j - \mathbf{w}\|_2^2] + \frac{G^2}{2} \sum_{j=1}^{t} (2\eta_{j-1}\eta_j + \eta_j^2) \\
&\leq \frac{1}{2} \|\mathbf{w}_0 - \mathbf{w}\|_2^2 + \frac{G^2}{2} \sum_{j=1}^{t} (2\eta_{j-1}\eta_j + \eta_j^2).
\end{aligned}
\tag{B.2}
$$

Taking the expectation on both sides of the above inequality and observing that $f(\cdot; z, z')$ is convex, we get the desired estimation (5).

For the strongly-convex case, i.e. $\alpha > 0$, we obtain from (B.1) that

$$
f(\mathbf{w}_{j-2}; z_{i_j}, z_{i_{j-1}}) - f(\mathbf{w}; z_{i_j}, z_{i_{j-1}}) \leq \frac{\eta_j^{-1} - \alpha}{2} \|\mathbf{w}_{j-1} - \mathbf{w}\|_2^2 - \frac{\eta_j^{-1}}{2} \|\mathbf{w}_j - \mathbf{w}\|_2^2 + G^2 \eta_{j-1} + \frac{G^2 \eta_j}{2}.
$$

Now, we choose $\eta_j = \frac{2}{\alpha(j+1)}$ for any $j$, which implies that

$$
\begin{aligned}
j[f(\mathbf{w}_{j-2}&; z_{i_j}, z_{i_{j-1}}) - f(\mathbf{w}; z_{i_j}, z_{i_{j-1}})] \\
&\leq \frac{j(j-1)\alpha}{4} \|\mathbf{w}_{j-1} - \mathbf{w}\|_2^2 - \frac{j(j+1)\alpha}{4} \|\mathbf{w}_j - \mathbf{w}\|_2^2 + \frac{2G^2}{\alpha} + \frac{G^2 j}{\alpha(j+1)} \\
&\leq \frac{\alpha}{4} [j(j-1)\|\mathbf{w}_{j-1} - \mathbf{w}\|_2^2 - j(j+1)\|\mathbf{w}_j - \mathbf{w}\|_2^2] + \frac{3G^2}{\alpha}.
\end{aligned}
$$

Taking the summation over $j$ implies that

$$
\begin{aligned}
\sum_{j=1}^{t} &j[f(\mathbf{w}_{j-2}; z_{i_j}, z_{i_{j-1}}) - f(\mathbf{w}; z_{i_j}, z_{i_{j-1}})] \\
&\leq \frac{3G^2 t}{\alpha} + \frac{\alpha}{4} \sum_{j=1}^{t} [j(j-1)\|\mathbf{w}_{j-1} - \mathbf{w}\|_2^2 - j(j+1)\|\mathbf{w}_j - \mathbf{w}\|_2^2] \\
&\leq \frac{3G^2 t}{\alpha} + \frac{\alpha}{4} [0 - t(t+1)\|\mathbf{w}_t - \mathbf{w}\|_2^2] \leq \frac{3G^2 t}{\alpha}.
\end{aligned}
\tag{B.3}
$$

Dividing both sides of the above inequality by $\sum_{j=1}^{t} j$ yields the desired estimation in part (b).  $\square$

To prove high-probability bounds, we require the following lemma on concentration inequalities of martingales [6, 47].

**Lemma 4.** *Let $\tilde{z}_1, \ldots, \tilde{z}_n$ be a sequence of random variables such that $\tilde{z}_k$ may depend on the previous variables $\tilde{z}_1, \ldots, \tilde{z}_{k-1}$ for all $k = 1, \ldots, n$. Consider a sequence of functionals $\xi_k(\tilde{z}_1, \ldots, \tilde{z}_k), k = 1, \ldots, n$. Let $\alpha_n^2 = \sum_{k=1}^{n} \mathbb{E}_{\tilde{z}_k}\left[\left(\xi_k - \mathbb{E}_{\tilde{z}_k}[\xi_k]\right)^2\right]$ be the conditional variance.*

*(1) Assume $|\xi_k - \mathbb{E}_{\tilde{z}_k}[\xi_k]| \leq b_k$ for each k. Let $\delta \in (0, 1)$. With probability at least $1 - \delta$*

$$\sum_{k=1}^{n} \mathbb{E}_{\tilde{z}_k}[\xi_k] - \sum_{k=1}^{n} \xi_k \leq \left(2 \sum_{k=1}^{n} b_k^2 \log \frac{1}{\delta}\right)^{\frac{1}{2}}. \tag{B.4}$$

*(2) Assume that $\xi_k - \mathbb{E}_{\tilde{z}_k}[\xi_k] \leq b$ for each k. Let $\rho \in (0, 1]$ and $\delta \in (0, 1)$. With probability at least $1 - \delta$ we have*

$$\sum_{k=1}^{n} \mathbb{E}_{\tilde{z}_k}[\xi_k] - \sum_{k=1}^{n} \xi_k \leq \frac{\rho \alpha_n^2}{b} + \frac{b \log \frac{1}{\delta}}{\rho}. \tag{B.5}$$

**Proof of Theorem 7.** For simplicity, we assume $t$ is an even number. We first consider the convex case. Let

$$\xi_j = \eta_{2j}\big(f(\mathbf{w}_{2j-2}; z_{i_{2j}}, z_{i_{2j-1}}) - f(\mathbf{w}; z_{i_{2j}}, z_{i_{2j-1}})\big), \quad j \in [t/2].$$

It is obvious that $|\xi_j - \mathbb{E}_{i_{2j}, i_{2j-1}}[\xi_j]| \leq 2B\eta_{2j}$. Let $\tilde{z}_j = (i_{2j}, i_{2j-1})$. It is clear that $\tilde{z}_j, j \in [t/2]$ are i.i.d. random variables. Therefore, one can apply Part (a) of Lemma 4 to derive the following inequality with probability at least $1 - \delta/2$

$$\sum_{j=1}^{t/2} \mathbb{E}_{\tilde{z}_j}[\xi_j] - \sum_{j=1}^{t/2} \xi_j \leq 2B\left(2 \sum_{j=1}^{t/2} \eta_{2j}^2 \log(2/\delta)\right)^{\frac{1}{2}}.$$

It is clear that $\mathbb{E}_{\tilde{z}_j}[\xi_j] = \eta_{2j}\big(F_S(\mathbf{w}_{2j-2}) - F_S(\mathbf{w})\big)$. Therefore, the following inequality holds with probability at least $1 - \delta/2$

$$\sum_{j=1}^{t/2} \eta_{2j}\big(F_S(\mathbf{w}_{2j-2}) - F_S(\mathbf{w}) - f(\mathbf{w}_{2j-2}; z_{i_{2j}}, z_{i_{2j-1}}) + f(\mathbf{w}; z_{i_{2j}}, z_{i_{2j-1}})\big) \leq 2B\left(2 \sum_{j=1}^{t/2} \eta_{2j}^2 \log(2/\delta)\right)^{\frac{1}{2}}.$$

In a similar way, one can derive the following inequality with probability at least $1 - \delta/2$

$$\sum_{j=1}^{t/2} \eta_{2j-1}\big(F_S(\mathbf{w}_{2j-3}) - F_S(\mathbf{w}) - f(\mathbf{w}_{2j-3}; z_{i_{2j-1}}, z_{i_{2j-2}})$$

$$+ f(\mathbf{w}; z_{i_{2j-1}}, z_{i_{2j-2}})\big) \leq 2B\left(2 \sum_{j=1}^{t/2} \eta_{2j-1}^2 \log(2/\delta)\right)^{\frac{1}{2}}.$$

We can combine the above two inequalities together and derive the following inequality with probability $1 - \delta$

$$\sum_{j=1}^{t} \eta_j\big(F_S(\mathbf{w}_{j-2}) - F_S(\mathbf{w}) - f(\mathbf{w}_{j-2}; z_{i_j}, z_{i_{j-1}}) + f(\mathbf{w}; z_{i_j}, z_{i_{j-1}})\big) \leq 2B\left(2 \sum_{j=1}^{t} \eta_j^2 \log(2/\delta)\right)^{\frac{1}{2}}.$$

We can combine the above inequality and Eq. (B.2) to derive the following inequality with probability at least $1 - \delta$

$$\sum_{j=1}^{t} \eta_j\big(F_S(\mathbf{w}_{j-2}) - F_S(\mathbf{w})\big) \leq 2B\left(2 \sum_{j=1}^{t} \eta_j^2 \log(2/\delta)\right)^{\frac{1}{2}} + \frac{1}{2}\|\mathbf{w}_0 - \mathbf{w}\|_2^2 + G^2 \sum_{j=1}^{t} \left(\eta_{j-1}\eta_j + \frac{\eta_j^2}{2}\right).$$

The stated bound then follows from the convexity of $F_S$.

We now turn to the strongly convex case. Let

$$\xi_j = 2j(f(\mathbf{w}_{2j-2}; z_{i_{2j}}, z_{i_{2j-1}}) - f(\mathbf{w}_S; z_{i_{2j}}, z_{i_{2j-1}})), \quad j \in [t/2].$$

It is clear that $|\xi_j - \mathbb{E}_{i_{2j}, i_{2j-1}}[\xi_j]| \leq 4jB \leq 2tB$ for $j \in [t/2]$. Furthermore, the conditional variance satisfies

$$\mathbb{E}_{\tilde{z}_j}\big[(\xi_j - \mathbb{E}_{\tilde{z}_j}[\xi_j])^2\big] \leq \mathbb{E}_{\tilde{z}_j}[\xi_j^2] \leq 4j^2 G^2 \|\mathbf{w}_{2j-2} - \mathbf{w}_S\|_2^2$$
$$\leq 8\alpha^{-1} j^2 G^2 \big(F_S(\mathbf{w}_{2j-2}) - F_S(\mathbf{w}_S)\big),$$

where the first inequality follows from $f$ is $G$-Lipschitz continuous, and the second inequality used the fact $\nabla F_S(\mathbf{w}_S) = 0$ and $f$ is $\alpha$-strongly convex.

Note $\tilde{z}_j$ are independent random variables and

$$\mathbb{E}_{\tilde{z}_j}[\xi_j] = 2j\big(F_S(\mathbf{w}_{2j-2}) - F_S(\mathbf{w}_S)\big).$$

Therefore, we can apply Part (b) of Lemma 4 to derive the following inequality with probability at least $1 - \delta/2$

$$2\sum_{j=1}^{t/2} j\Big(F_S(\mathbf{w}_{2j-2}) - F_S(\mathbf{w}_S) - f(\mathbf{w}_{2j-2}; z_{i_{2j}}, z_{i_{2j-1}}) + f(\mathbf{w}_S; z_{i_{2j}}, z_{i_{2j-1}})\Big)$$
$$\leq \frac{8G^2 \rho \sum_{j=1}^{t/2} j^2 \big(F_S(\mathbf{w}_{2j-2}) - F_S(\mathbf{w}_S)\big)}{2tB\alpha} + \frac{2tB \log(2/\delta)}{\rho}.$$

In a similar way, one can derive the following inequality with probability at least $1 - \delta/2$

$$\sum_{j=1}^{t/2} (2j-1)\Big(F_S(\mathbf{w}_{2j-3}) - F_S(\mathbf{w}_S) - f(\mathbf{w}_{2j-3}; z_{i_{2j-1}}, z_{i_{2j-2}}) + f(\mathbf{w}_S; z_{i_{2j-1}}, z_{i_{2j-2}})\Big)$$
$$\leq \frac{2G^2 \rho \sum_{j=1}^{t/2} (2j-1)^2 \big(F_S(\mathbf{w}_{2j-3}) - F_S(\mathbf{w}_S)\big)}{2tB\alpha} + \frac{2tB \log(2/\delta)}{\rho}.$$

We can combine the above two inequalities together and derive the following inequality with probability at least $1 - \delta$

$$\sum_{j=1}^{t} j\Big(F_S(\mathbf{w}_{j-2}) - F_S(\mathbf{w}_S) - f(\mathbf{w}_{j-2}; z_{i_j}, z_{i_{j-1}}) + f(\mathbf{w}_S; z_{i_j}, z_{i_{j-1}})\Big)$$
$$\leq \frac{2G^2 \rho \sum_{j=1}^{t} j^2 \big(F_S(\mathbf{w}_{j-2}) - F_S(\mathbf{w}_S)\big)}{2tB\alpha} + \frac{2tB \log(2/\delta)}{\rho}.$$

We can combine the above inequality and Eq. (B.3) together and derive the following inequality with probability $1 - \delta$

$$\sum_{j=1}^{t} j\big(F_S(\mathbf{w}_{j-2}) - F_S(\mathbf{w}_S)\big) \leq \frac{3G^2 t}{\alpha} + \frac{G^2 \rho \sum_{j=1}^{t} j\big(F_S(\mathbf{w}_{j-2}) - F_S(\mathbf{w}_S)\big)}{B\alpha} + \frac{2tB \log(2/\delta)}{\rho}.$$

Now, we take $\rho = \min\big\{1, B\alpha/(2G^2)\big\}$ and get the following inequality with probability at least $1 - \delta$

$$\sum_{j=1}^{t} j\big(F_S(\mathbf{w}_{j-2}) - F_S(\mathbf{w}_S)\big) \leq \frac{3G^2 t}{\alpha} + \frac{1}{2}\sum_{j=1}^{t} j\big(F_S(\mathbf{w}_{j-2}) - F_S(\mathbf{w}_S)\big) + 2t \log(2/\delta) \max\big\{B, 2G^2/\alpha\big\}$$

and therefore

$$\sum_{j=1}^{t} j\big(F_S(\mathbf{w}_{j-2}) - F_S(\mathbf{w}_S)\big) \leq \frac{14G^2 t \log(2/\delta)}{\alpha} + 4Bt \log(2/\delta).$$

The stated bound then follows from the convexity of $F_S$. The proof is completed. $\qquad\square$

We now turn to nonconvex problems.

**Proof of Theorem 8.** It is clear that $F_S$ is $L$-smooth and therefore

$$F_S(\mathbf{w}_j) \leq F_S(\mathbf{w}_{j-1}) + \langle \mathbf{w}_j - \mathbf{w}_{j-1}, \nabla F_S(\mathbf{w}_{j-1}) \rangle + \frac{L}{2} \|\mathbf{w}_j - \mathbf{w}_{j-1}\|_2^2$$

$$= F_S(\mathbf{w}_{j-1}) - \eta_j \langle \nabla f(\mathbf{w}_{j-1}; z_{i_j}, z_{i_{j-1}}), \nabla F_S(\mathbf{w}_{j-1}) \rangle + \frac{L\eta_j^2}{2} \|\nabla f(\mathbf{w}_{j-1}; z_{i_j}, z_{i_{j-1}})\|_2^2$$

Taking expectations over both sides gives

$$\mathbb{E}_{\mathcal{A}}[F_S(\mathbf{w}_j)] \leq \mathbb{E}_{\mathcal{A}}[F_S(\mathbf{w}_{j-1})] - \eta_j \mathbb{E}_{\mathcal{A}}\big[\langle \nabla f(\mathbf{w}_{j-1}; z_{i_j}, z_{i_{j-1}}), \nabla F_S(\mathbf{w}_{j-1}) \rangle\big] +$$
$$\frac{L\eta_j^2}{2} \mathbb{E}_{\mathcal{A}}\big[\|\nabla f(\mathbf{w}_{j-1}; z_{i_j}, z_{i_{j-1}})\|_2^2\big]. \quad \text{(B.6)}$$

According to the elementary inequality $(a+b)^2 \leq 2(a^2 + b^2)$ we know

$$\mathbb{E}_{\mathcal{A}}\big[\|\nabla f(\mathbf{w}_{j-1}; z_{i_j}, z_{i_{j-1}})\|_2^2\big]$$
$$\leq 2\mathbb{E}_{\mathcal{A}}\big[\|\nabla f(\mathbf{w}_{j-1}; z_{i_j}, z_{i_{j-1}}) - \nabla f(\mathbf{w}_{j-2}; z_{i_j}, z_{i_{j-1}})\|_2^2\big] + 2\mathbb{E}_{\mathcal{A}}\big[\|\nabla f(\mathbf{w}_{j-2}; z_{i_j}, z_{i_{j-1}})\|_2^2\big]$$
$$\leq 2L\mathbb{E}_{\mathcal{A}}\big[\|\mathbf{w}_{j-1} - \mathbf{w}_{j-2}\|_2^2\big] + 2\alpha_0^2 = 2L\eta_{j-1}^2 \mathbb{E}_{\mathcal{A}}\big[\|\nabla f(\mathbf{w}_{j-2}; z_{i_{j-1}}, z_{i_{j-2}})\|_2^2\big] + 2\alpha_0^2$$
$$\leq \frac{1}{2}\mathbb{E}_{\mathcal{A}}\big[\|\nabla f(\mathbf{w}_{j-2}; z_{i_{j-1}}, z_{i_{j-2}})\|_2^2\big] + 2\alpha_0^2,$$

where we have used the $L$-smoothness, the assumption $\mathbb{E}_{i_j, i_{j-1}}\big[\|\nabla f(\mathbf{w}_{j-2}; z_{i_j}, z_{i_{j-1}})\|_2^2\big] \leq \alpha_0^2$ and $4L\eta_{j-1}^2 \leq 1$. It is clear that $\mathbb{E}_{\mathcal{A}}\big[\|\nabla f(\mathbf{w}_0; z_{i_1}, z_{i_0})\|_2^2\big] \leq \alpha_0^2$. It is easy to use an induction and the above inequality to show that

$$\mathbb{E}_{\mathcal{A}}\big[\|\nabla f(\mathbf{w}_{j-1}; z_{i_j}, z_{i_{j-1}})\|_2^2\big] \leq 4\alpha_0^2, \quad \forall j. \quad \text{(B.7)}$$

Furthermore, the smoothness assumption implies that

$$\langle \nabla f(\mathbf{w}_{j-1}; z_{i_j}, z_{i_{j-1}}), \nabla F_S(\mathbf{w}_{j-1}) \rangle$$
$$= \langle \nabla f(\mathbf{w}_{j-2}; z_{i_j}, z_{i_{j-1}}), \nabla F_S(\mathbf{w}_{j-1}) \rangle + \langle \nabla f(\mathbf{w}_{j-1}; z_{i_j}, z_{i_{j-1}}) - \nabla f(\mathbf{w}_{j-2}; z_{i_j}, z_{i_{j-1}}), \nabla F_S(\mathbf{w}_{j-1}) \rangle$$
$$= \langle \nabla f(\mathbf{w}_{j-2}; z_{i_j}, z_{i_{j-1}}), \nabla F_S(\mathbf{w}_{j-2}) \rangle + \langle \nabla f(\mathbf{w}_{j-2}; z_{i_j}, z_{i_{j-1}}), \nabla F_S(\mathbf{w}_{j-1}) - \nabla F_S(\mathbf{w}_{j-2}) \rangle +$$
$$+ \langle \nabla f(\mathbf{w}_{j-1}; z_{i_j}, z_{i_{j-1}}) - \nabla f(\mathbf{w}_{j-2}; z_{i_j}, z_{i_{j-1}}), \nabla F_S(\mathbf{w}_{j-1}) \rangle$$
$$\geq \langle \nabla f(\mathbf{w}_{j-2}; z_{i_j}, z_{i_{j-1}}), \nabla F_S(\mathbf{w}_{j-2}) \rangle - L\|\mathbf{w}_{j-1} - \mathbf{w}_{j-2}\|_2\big(\|\nabla f(\mathbf{w}_{j-2}; z_{i_j}, z_{i_{j-1}})\|_2 + \|\nabla F_S(\mathbf{w}_{j-1})\|_2\big).$$

According to Schwartz inequality, the variance assumption and Eq. (B.7), we know

$$\mathbb{E}_{\mathcal{A}}\Big[\|\mathbf{w}_{j-1} - \mathbf{w}_{j-2}\|_2\big(\|\nabla f(\mathbf{w}_{j-2}; z_{i_j}, z_{i_{j-1}})\|_2 + \|\nabla F_S(\mathbf{w}_{j-1})\|_2\big)\Big]$$
$$\leq \frac{1}{2\eta_{j-1}}\mathbb{E}_{\mathcal{A}}\big[\|\mathbf{w}_{j-1} - \mathbf{w}_{j-2}\|_2^2\big] + \eta_{j-1}\mathbb{E}_{\mathcal{A}}\big[\|\nabla f(\mathbf{w}_{j-2}; z_{i_j}, z_{i_{j-1}})\|_2^2 + \|\nabla F_S(\mathbf{w}_{j-1})\|_2^2\big]$$
$$= \frac{\eta_{j-1}}{2}\mathbb{E}_{\mathcal{A}}\big[\|\nabla f(\mathbf{w}_{j-2}; z_{i_{j-1}}, z_{i_{j-2}})\|_2^2\big] + \eta_{j-1}\mathbb{E}_{\mathcal{A}}\big[\|\nabla f(\mathbf{w}_{j-2}; z_{i_j}, z_{i_{j-1}})\|_2^2 + \|\nabla F_S(\mathbf{w}_{j-1})\|_2^2\big]$$
$$\leq 2\alpha_0^2\eta_{j-1} + 2\alpha_0^2\eta_{j-1}.$$

We can combine the above two inequalities together and get

$$\mathbb{E}_{\mathcal{A}}\big[\langle \nabla f(\mathbf{w}_{j-1}; z_{i_j}, z_{i_{j-1}}), \nabla F_S(\mathbf{w}_{j-1}) \rangle\big] \geq \mathbb{E}_{\mathcal{A}}\big[\langle \nabla f(\mathbf{w}_{j-2}; z_{i_j}, z_{i_{j-1}}), \nabla F_S(\mathbf{w}_{j-2}) \rangle\big] - 4L\alpha_0^2\eta_{j-1}.$$

We can combine (B.6), (B.7) and the above inequality, and get

$$\mathbb{E}_{\mathcal{A}}[F_S(\mathbf{w}_j)] \leq \mathbb{E}_{\mathcal{A}}[F_S(\mathbf{w}_{j-1})] - \eta_j \mathbb{E}_{\mathcal{A}}\big[\langle \nabla f(\mathbf{w}_{j-2}; z_{i_j}, z_{i_{j-1}}), \nabla F_S(\mathbf{w}_{j-2}) \rangle\big] + 4L\alpha_0^2\eta_j\eta_{j-1} + 2L\eta_j^2\alpha_0^2$$
$$\leq \mathbb{E}_{\mathcal{A}}[F_S(\mathbf{w}_{j-1})] - \eta_j \mathbb{E}_{\mathcal{A}}\big[\|\nabla F_S(\mathbf{w}_{j-2})\|_2^2\big] + 4L\alpha_0^2\big(\eta_j\eta_{j-1} + \eta_j^2\big),$$

where the last inequality holds since $\mathbf{w}_{j-2}$ is independent of $i_j$ and $i_{j-1}$. The above inequality can be reformulated as

$$\eta_j \mathbb{E}_{\mathcal{A}}\big[\|\nabla F_S(\mathbf{w}_{j-2})\|_2^2\big] \leq \mathbb{E}_{\mathcal{A}}[F_S(\mathbf{w}_{j-1})] - \mathbb{E}_{\mathcal{A}}[F_S(\mathbf{w}_j)] + 4L\alpha_0^2\big(\eta_j\eta_{j-1} + \eta_j^2\big). \quad \text{(B.8)}$$

We can take a summation of the above inequality and get

$$\sum_{j=1}^{t} \eta_j \mathbb{E}_{\mathcal{A}}\big[\|\nabla F_S(\mathbf{w}_{j-2})\|_2^2\big] \leq F_S(\mathbf{w}_0) + 4L\alpha_0^2 \sum_{j=1}^{t} \big(\eta_j\eta_{j-1} + \eta_j^2\big).$$

Since $\eta_j = \eta$, we further get

$$\sum_{j=1}^{t} \mathbb{E}_{\mathcal{A}}\left[\|\nabla F_S(\mathbf{w}_{j-2})\|_2^2\right] \leq \eta^{-1} F_S(\mathbf{w}_0) + 8L\alpha_0^2 t\eta.$$

The proof is completed. $\qquad\square$

**Proof of Theorem 9.** According to the elementary inequality $\frac{1}{2}(a+b)^2 \leq a^2 + b^2$ we know

$$\begin{aligned}
\mathbb{E}_{\mathcal{A}}\left[\|\nabla F_S(\mathbf{w}_{j-2})\|_2^2\right] &\geq -\mathbb{E}_{\mathcal{A}}\left[\|\nabla F_S(\mathbf{w}_{j-2}) - \nabla F_S(\mathbf{w}_{j-1})\|_2^2\right] + 2^{-1}\mathbb{E}_{\mathcal{A}}\left[\|\nabla F_S(\mathbf{w}_{j-1})\|_2^2\right] \\
&\geq -L\mathbb{E}_{\mathcal{A}}\left[\|\mathbf{w}_{j-2} - \mathbf{w}_{j-1}\|_2^2\right] + 2^{-1}\mathbb{E}_{\mathcal{A}}\left[\|\nabla F_S(\mathbf{w}_{j-1})\|_2^2\right] \\
&= -L\eta_{j-1}^2\mathbb{E}_{\mathcal{A}}\left[\|\nabla f(\mathbf{w}_{j-2}; z_{i_{j-1}}, z_{i_{j-2}})\|_2^2\right] + 2^{-1}\mathbb{E}_{\mathcal{A}}\left[\|\nabla F_S(\mathbf{w}_{j-1})\|_2^2\right] \\
&\geq -4L\eta_{j-1}^2\alpha_0^2 + 2^{-1}\mathbb{E}_{\mathcal{A}}\left[\|\nabla F_S(\mathbf{w}_{j-1})\|_2^2\right],
\end{aligned}$$

where we have used (B.7). This together with (B.8) gives

$$2^{-1}\eta_j\mathbb{E}_{\mathcal{A}}\left[\|\nabla F_S(\mathbf{w}_{j-1})\|_2^2\right] \leq 4L\eta_j\eta_{j-1}^2\alpha_0^2 + \mathbb{E}_{\mathcal{A}}[F_S(\mathbf{w}_{j-1})] - \mathbb{E}_{\mathcal{A}}[F_S(\mathbf{w}_j)] + 4L\alpha_0^2\big(\eta_j\eta_{j-1} + \eta_j^2\big).$$

It then follows from the PL condition that

$$\mu\eta_j\mathbb{E}_{\mathcal{A}}\left[F_S(\mathbf{w}_{j-1}) - F_S(\mathbf{w}_S)\right] \leq \mathbb{E}_{\mathcal{A}}[F_S(\mathbf{w}_{j-1})] - \mathbb{E}_{\mathcal{A}}[F_S(\mathbf{w}_j)] + 4L\alpha_0^2\big(\eta_j\eta_{j-1} + \eta_j^2 + \eta_j\eta_{j-1}^2\big).$$

We can reformulate the above inequality as

$$\mathbb{E}_{\mathcal{A}}[F_S(\mathbf{w}_j) - F_S(\mathbf{w}_S)] \leq (1 - \mu\eta_j)\mathbb{E}_{\mathcal{A}}[F_S(\mathbf{w}_{j-1}) - F_S(\mathbf{w}_S)] + 4L\alpha_0^2\big(\eta_j\eta_{j-1} + \eta_j^2 + \eta_j\eta_{j-1}^2\big).$$

Now, taking $\eta_j = 2/(\mu(j+1))$, we get

$$\mathbb{E}_{\mathcal{A}}[F_S(\mathbf{w}_j) - F_S(\mathbf{w}_S)] \leq \frac{j-1}{j+1}\mathbb{E}_{\mathcal{A}}[F_S(\mathbf{w}_{j-1}) - F_S(\mathbf{w}_S)] + \frac{4L\alpha_0^2}{\mu^2}\Big(\frac{8}{j(j+1)} + \frac{8}{j^2(j+1)\mu}\Big).$$

We can multiple both sides by $j(j+1)$ and get

$$j(j+1)\mathbb{E}_{\mathcal{A}}[F_S(\mathbf{w}_j) - F_S(\mathbf{w}_S)] \leq (j-1)j\mathbb{E}_{\mathcal{A}}[F_S(\mathbf{w}_{j-1}) - F_S(\mathbf{w}_S)] + \frac{4L\alpha_0^2}{\mu^2}\big(8 + 8j^{-1}\mu^{-1}\big).$$

Taking a summation of the above inequality gives

$$t(t+1)\mathbb{E}_{\mathcal{A}}[F_S(\mathbf{w}_t) - F_S(\mathbf{w}_S)] \leq \frac{32L\alpha_0^2}{\mu^2}\sum_{j=1}^{t}\big(1 + j^{-1}\mu^{-1}\big).$$

The stated bound then follows. The proof is completed. $\qquad\square$

## C  Proofs of Excess Generalization Error (Theorems 1-3)

In this section, we prove Theorem 1, Theorem 2 and Theorem 3 on excess generalization error bounds.

**Proof of Theorem 1.** According to Theorem 4, we know

$$\mathbb{E}_{S,\mathcal{A}}[F(\bar{\mathbf{w}}_T) - F_S(\bar{\mathbf{w}}_T)] = \mathcal{O}\Big(\sqrt{T}\eta + \frac{T\eta}{n}\Big).$$

Furthermore, according to Part (a) of Theorem 6 with $\mathbf{w} = \mathbf{w}^*$, we know

$$\mathbb{E}_{S,\mathcal{A}}[F_S(\bar{\mathbf{w}}_T) - F_S(\mathbf{w}^*)] = \mathcal{O}\Big(\frac{1 + T\eta^2}{T\eta}\Big). \tag{C.1}$$

We can plug the above generalization error bound and optimization error bound back into the error decomposition (2), and get (3). Taking $T \asymp n^2$ and $\eta \asymp T^{-\frac{3}{4}}$ in Eq. (3), we immediately get $\mathbb{E}_{S,\mathcal{A}}[F(\bar{\mathbf{w}}_T)] - F(\mathbf{w}^*) = \mathcal{O}(1/\sqrt{n})$. The desired result is proved. $\qquad\square$

**Proof of Theorem 2.** According to Theorem 5, we know

$$\mathbb{E}_{S,\mathcal{A}}[F(\bar{\mathbf{w}}_T) - F_S(\bar{\mathbf{w}}_T)] = \mathcal{O}\Big(\frac{T\eta}{n}\Big).$$

We can plug the above generalization error bound and the optimization error bound (C.1) back into the error decomposition (2), and get (4). Taking $T \asymp n^2$ and $\eta \asymp T^{-\frac{3}{4}}$ in Eq. (4), we immediately get $\mathbb{E}_{S,\mathcal{A}}[F(\bar{\mathbf{w}}_T)] - F(\mathbf{w}^*) = \mathcal{O}(1/\sqrt{n})$. The proof is completed. $\qquad\square$

**Proof of Theorem 3.** Let $S' = \{z_1, \ldots, z_{n-1}, z_n'\}$, where $z_n'$ is independently drawn from $\rho$. Let $\{\mathbf{w}_t'\}$ be produced by Algorithm 1 w.r.t. $S'$. If $i_t \neq n$ and $i_{t-1} \neq n$, then

$$
\begin{aligned}
\|\mathbf{w}_t - \mathbf{w}_t'\|_2 &\leq \big\|\mathbf{w}_{t-1} - \eta_t\nabla f(\mathbf{w}_{t-1}; z_{i_t}, z_{i_{t-1}}) - \mathbf{w}_{t-1}' - \eta_t\nabla f(\mathbf{w}_{t-1}'; z_{i_t}, z_{i_{t-1}})\big\|_2 \\
&\leq \|\mathbf{w}_{t-1} - \mathbf{w}_{t-1}'\|_2 + \eta_t\|\nabla f(\mathbf{w}_{t-1}; z_{i_t}, z_{i_{t-1}}) - \nabla f(\mathbf{w}_{t-1}'; z_{i_t}, z_{i_{t-1}})\|_2 \\
&\leq (1 + L\eta_t)\|\mathbf{w}_{t-1} - \mathbf{w}_{t-1}'\|_2,
\end{aligned}
$$

where in the last inequality we used the smoothness of $f$.

Otherwise, it follows from the Lipschitz condition that $\|\mathbf{w}_t - \mathbf{w}_t'\|_2 \leq \|\mathbf{w}_{t-1} - \mathbf{w}_{t-1}'\|_2 + 2G\eta_t$. Consequently, it follows that

$$
\begin{aligned}
\|\mathbf{w}_t - \mathbf{w}_t'\|_2 &\leq (1 + L\eta_t)\|\mathbf{w}_{t-1} - \mathbf{w}_{t-1}'\|_2 \mathbb{I}_{[i_t \neq n \text{ and } i_{t-1} \neq n]} \\
&\quad + \big(\|\mathbf{w}_{t-1} - \mathbf{w}_{t-1}'\|_2 + 2G\eta_t\big)\mathbb{I}_{[i_t = n \text{ or } i_{t-1} = n]} \\
&\leq (1 + L\eta_t)\|\mathbf{w}_{t-1} - \mathbf{w}_{t-1}'\|_2 + 2G\eta_t\mathbb{I}_{[i_t = n \text{ or } i_{t-1} = n]}.
\end{aligned}
$$

We can apply the above inequality recursively and get

$$\triangle_t \leq 2G\sum_{k=t_0+1}^{t} \eta_k\mathbb{I}_{[i_k=n \text{ or } i_{k-1}=n]} \prod_{k'=k+1}^{t}(1 + L\eta_{k'}) + \triangle_{t_0}\prod_{k=t_0+1}^{t}(1 + L\eta_k).$$

Since $\triangle_{t_0} = 0$ implies $i_{t_0} \neq n$, we have

$$
\begin{aligned}
\mathbb{E}[\triangle_t|\triangle_{t_0} = 0] &\leq 2G\sum_{k=t_0+1}^{t} \eta_k\mathbb{E}\big[\mathbb{I}_{[i_k=n \text{ or } i_{k-1}=n]}|\triangle_{t_0} = 0\big]\prod_{k'=k+1}^{t}(1 + L\eta_{k'}) \\
&= 2G\sum_{k=t_0+2}^{t} \eta_k\mathbb{E}\big[\mathbb{I}_{[i_k=n \text{ or } i_{k-1}=n]}|\triangle_{t_0} = 0\big]\prod_{k'=k+1}^{t}(1 + L\eta_{k'}) \\
&= 2G\sum_{k=t_0+2}^{t} \eta_k\mathbb{E}\big[\mathbb{I}_{[i_k=n \text{ or } i_{k-1}=n]}\big]\prod_{k'=k+1}^{t}(1 + L\eta_{k'}),
\end{aligned}
$$

where we have used the independency between $\triangle_{t_0}$ and $i_t$ for $t > t_0$. It then follows that

$$
\begin{aligned}
\mathbb{E}[\triangle_t|\triangle_{t_0} = 0] &\leq 2G\sum_{k=t_0+2}^{t} \eta_k\mathbb{E}\big[\mathbb{I}_{[i_k=n]} + \mathbb{I}_{[i_{k-1}=n]}\big]\prod_{k'=k+1}^{t}(1 + L\eta_{k'}) \\
&\leq \frac{4G}{n}\sum_{k=t_0+2}^{t} \eta_k\prod_{k'=k+1}^{t}\exp(L\eta_{k'}) \leq \frac{8G}{\mu n}\sum_{k=t_0+2}^{t}\frac{1}{k+1}\exp\Big(2L\mu^{-1}\sum_{k'=k+1}^{t}\frac{1}{k'+1}\Big) \\
&\leq \frac{8G}{\mu n}\sum_{k=t_0+2}^{t}\frac{1}{k+1}\exp\Big(2L\mu^{-1}\log(t/k)\Big) = \frac{8G}{\mu n}\sum_{k=t_0+2}^{t}\frac{1}{k+1}\Big(\frac{t}{k}\Big)^{2L\mu^{-1}} \\
&\leq \frac{8Gt^{2L\mu^{-1}}}{\mu n}\sum_{k=t_0+2}^{t} k^{-1-2L\mu^{-1}} \leq \frac{8G}{\mu n(2L\mu^{-1})}\Big(\frac{t}{t_0}\Big)^{2L\mu^{-1}}.
\end{aligned}
$$

Here we use $1 + x \leq \exp(x)$ and $\eta_j = \frac{2}{\mu(j+1)}$. We can plug the above inequality back into Lemma 3 and derive

$$\mathbb{E}\big[|f(\mathbf{w}_T; z, z') - f(\mathbf{w}_T'; z, z')|\big] \leq \frac{4G^2}{nL}\Big(\frac{T}{t_0}\Big)^{2L\mu^{-1}} + \frac{Bt_0}{n}.$$

We can choose $t_0 \asymp T^{\frac{2L}{2L+\mu}}$ and get the following generalization error bounds

$$\mathbb{E}\big[|f(\mathbf{w}_T; z, z') - f(\mathbf{w}_T'; z, z')|\big] = \mathcal{O}\Big(\frac{T^{\frac{2L}{2L+\mu}}}{n}\Big).$$

Lemma 1 then implies $\mathbb{E}\big[F(\mathbf{w}_T) - F_S(\mathbf{w}_T)\big] = \mathcal{O}\Big(\frac{T^{\frac{2L}{2L+\mu}}}{n}\Big)$. Furthermore, according to Theorem 9 we have the following optimization error bounds

$$\mathbb{E}_A[F_S(\mathbf{w}_T)] - \inf_{\mathbf{w}}[F_S(\mathbf{w})] = \mathcal{O}\big(1/(T\mu^2)\big).$$

The desired result follows by combining the above two inequalities together and using the fact $\mathbb{E}[\inf_{\mathbf{w}}[F_S(\mathbf{w})] \leq \mathbb{E}[F_S(\mathbf{w}^*)] = F(\mathbf{w}^*)$. $\qquad\square$

# D   Proof of Privacy and Utility Guarantees of Algorithm 3 (Theorem 10)

In this section, we present the proof of Theorem 10 on the privacy guarantee and excess generalization bound of Algorithm 3.

To this end, we need the definition of $\ell_2$-sensitivity in terms of high probability and some lemmas. The $\ell_2$-sensitivity definition given below corresponds to the high probability version of uniform argument stability stated in Definition 1.

**Definition 3.** For any $\gamma \in (0, 1)$, a (randomized) algorithm $\mathcal{A}$ has $\ell_2$-sensitivity of $\Delta$ with probability at least $1 - \gamma$ if for any neighboring datasets $S, S' \in \mathcal{Z}^n$, one has $\|\mathcal{A}(S) - \mathcal{A}(S')\|_2 \leq \Delta$.

The next lemma demonstrates that Gaussian mechanism ensures the privacy of an algorithm with high probability $\ell_2$-sensitivity.

**Lemma 5.** *Let $\mathcal{A} : \mathcal{Z}^n \to \mathbb{R}^d$ be a randomized algorithm with $\ell_2$-sensitivity of $\Delta$ with probability at least $1 - \delta/2$. Then the Gaussian mechanism $\mathcal{M}(S) = \mathcal{A}(S) + \mathbf{u}$ where $\mathbf{u} \sim \mathcal{N}(0, (2\Delta^2 \log(2.5/\delta)/\epsilon^2)I_d)$ satisfies $(\epsilon, \delta)$-DP.*

*Proof.* Let $S$ and $S'$ be two neighboring datasets. Denote $E$ as the set when $\mathcal{A}$ satisfies $\ell_2$-sensitivity of $\Delta$, i.e. $E = \{\|\mathcal{A}(S) - \mathcal{A}(S')\|_2 \leq \Delta\}$. Then we know $\mathbb{P}[E] \geq 1 - \delta/2$. In favor of $E$, by classical results for Gaussian mechanism, we know $\mathcal{M}$ satisfies $(\epsilon, \delta/2)$-DP with $\sigma = \Delta\sqrt{2\log(2.5/\delta)}/\epsilon$. Therefore, for any $\epsilon > 0$ and any event $O$ in the output space of $\mathcal{M}$, we have

$$\begin{aligned}
\mathbb{P}[\mathcal{M}(S) \in O] &= \mathbb{P}[\mathcal{M}(S) \in O|E]\mathbb{P}[E] + \mathbb{P}[\mathcal{M}(S) \in O|E^c]\mathbb{P}[E^c] \\
&\leq \Big(e^\epsilon \mathbb{P}[\mathcal{M}(S') \in O|E] + \frac{\delta}{2}\Big)\mathbb{P}[E] + \frac{\delta}{2} \\
&\leq e^\epsilon \mathbb{P}[\mathcal{M}(S') \in O \cap E] + \frac{\delta}{2} + \frac{\delta}{2} \\
&\leq e^\epsilon \mathbb{P}[\mathcal{M}(S') \in O] + \delta
\end{aligned}$$

where the first inequality is because $\mathcal{M}$ satisfies $(\epsilon, \delta/2)$-DP when $E$ occurs and $\mathbb{P}[E^c] \leq \delta/2$, the second and third inequalities are by basic properties of probability. The proof is completed. $\qquad\square$

We need the following Chernoff's bound for a summation of independent Bernoulli random variables [43].

**Lemma 6** (Chernoff bound for Bernoulli vector). *Let $X_1, \ldots, X_t$ be independent random variables taking values in $\{0, 1\}$. Let $X = \sum_{j=1}^t X_j$ and $\mu = \mathbb{E}[X]$. Then for any $\tilde{\gamma} > 0$, with probability at least $1 - \exp\big(-\mu\tilde{\gamma}^2/(2 + \tilde{\gamma})\big)$ we have $X \leq (1 + \tilde{\gamma})\mu$.*

In order to prove the privacy guarantee and excess generalization bound for Algorithm 3, we also need the following high probability $\ell_2$-sensitivity of the output of Algorithm 1.

**Lemma 7.** *Let $\{\bar{\mathbf{w}}_t\}$ and $\{\bar{\mathbf{w}}_t'\}$ be produced by Algorithm 1 based on the neighboring datasets $S$ and $S'$, respectively. If $f$ is convex and $L$-smooth and $\eta_t = \eta \leq 2/L$, then with probability at least $1 - \gamma$ we have*

$$\|\bar{\mathbf{w}}_t - \bar{\mathbf{w}}_t'\|_2 \leq 4G\eta\Big(\frac{t}{n} + \log(2/\gamma) + \sqrt{\frac{t\log(2/\gamma)}{n}}\Big).$$

*Proof.* Without loss of generality, we assume the different example between $S$ and $S'$ is the $n$-th item. By the proof of Theorem 5, we know

$$\left\|\mathbf{w}_t - \mathbf{w}'_t\right\|_2 \leq 2G \sum_{j=1}^{t} \eta_j \mathbb{I}_{[i_j=n \text{ or } i_{j-1}=n]} \leq 2G \sum_{j=1}^{t} \eta_j \left(\mathbb{I}_{[i_j=n]} + \mathbb{I}_{[i_{j-1}=n]}\right).$$

Applying Lemma 6, with probability at least $1 - \gamma$ there holds

$$\sum_{j=1}^{t} (\mathbb{I}_{[i_j=n]} + \mathbb{I}_{[i_{j-1}=n]}) \leq \frac{2t}{n} + 2\log(2/\gamma) + 2\sqrt{\frac{t\log(2/\gamma)}{n}}.$$

It then follows from the convexity of $\|\cdot\|_2$ that

$$\left\|\bar{\mathbf{w}}_t - \bar{\mathbf{w}}'_t\right\|_2 \leq 4G\eta\left(\frac{t}{n} + \log(2/\gamma) + \sqrt{\frac{t\log(2/\gamma)}{n}}\right),$$

which implies the desired result. $\qquad\square$

With the above preparations, we are now ready to prove Theorem 10.

**Proof of Theorem 10.** We first consider the privacy guarantee of Algorithm 3. Since we run Algorithm 1 for $\lceil n_k \log(4/\delta) \rceil$ steps for each $k$, by Lemma 7, we know with probability $1 - \delta/2$

$$\left\|\bar{\mathbf{w}}_k - \bar{\mathbf{w}}'_k\right\|_2 \leq 12G\eta_k \log(4/\delta).$$

Therefore, by Lemma 5, each iteration $k$ of Algorithm 3 is $(\epsilon, \delta)$-DP. Since the partition of the dataset $S$ is disjoint, and each iteration $k$ of Algorithm 3 we only use one subset, thus the whole process satisfies $(\epsilon, \delta)$-DP.

Next we investigate the utility bound of Algorithm 3. Let $\bar{\mathbf{w}}_0 = \mathbf{w}^*$ and $\mathbf{u}_0 = \mathbf{w}_0 - \mathbf{w}^*$, then

$$\mathbb{E}[F(\mathbf{w}_K) - F(\mathbf{w}^*)] = \sum_{k=1}^{K} \mathbb{E}[F(\bar{\mathbf{w}}_k) - F(\bar{\mathbf{w}}_{k-1})] + \mathbb{E}[F(\mathbf{w}_K) - F(\bar{\mathbf{w}}_K)] \qquad \text{(D.1)}$$

Denote $F_{S_k}$ be the empirical objective based on sample $S_k$. For the first term on the RHS of (D.1), we have

$$\begin{aligned}
&\mathbb{E}[F(\bar{\mathbf{w}}_k) - F(\bar{\mathbf{w}}_{k-1})] \\
&= \mathbb{E}[F(\bar{\mathbf{w}}_k) - F_{S_k}(\bar{\mathbf{w}}_k)] + \mathbb{E}[F_{S_k}(\bar{\mathbf{w}}_k) - F_{S_k}(\bar{\mathbf{w}}_{k-1})] + \mathbb{E}[F_{S_k}(\bar{\mathbf{w}}_{k-1}) - F(\bar{\mathbf{w}}_{k-1})] \\
&= \mathbb{E}[F(\bar{\mathbf{w}}_k) - F_{S_k}(\bar{\mathbf{w}}_k)] + \mathbb{E}[F_{S_k}(\bar{\mathbf{w}}_k) - F_{S_k}(\bar{\mathbf{w}}_{k-1})] \\
&\leq 8G^2 \log(4/\delta)\eta_k + \left(\frac{\mathbb{E}[\|\mathbf{u}_{k-1}\|_2^2]}{2\eta_k n_k} + \frac{3G^2\eta_k}{2}\right) \leq \frac{\mathbb{E}[\|\mathbf{u}_{k-1}\|_2^2]}{2\eta_k n_k} + 18\log(4/\delta)G^2\eta_k,
\end{aligned}$$

where the second identity is because $\bar{\mathbf{w}}_{k-1}$ is independent of $S_k$ and the inequality follows from Theorem 6 Part (a) and Theorem 5. Recall that by definition $\eta \leq \frac{D\epsilon}{12G\log(4/\delta)\sqrt{2d\log(2.5/\delta)}}$, so that for all $k \geq 0$,

$$\mathbb{E}[\|\mathbf{u}_k\|_2^2] = d\sigma_k^2 = d\left(\frac{4^{-k}G\eta}{\epsilon}\right)^2 \leq 16^{-k}D^2.$$

Plugging the above estimate into (D.1) it follows

$$\begin{aligned}
\mathbb{E}[F(\mathbf{w}_K) - F(\mathbf{w}^*)] &\leq \sum_{k=1}^{K} \frac{8 \cdot 16^{-k}D^2}{4^{-k}2^{-k}\eta n} + 18\log(4/\delta)4^{-k}G^2\eta + 4^{-K}GD \\
&\leq \sum_{k=1}^{K} 2^{-k}\left(\frac{8D^2}{\eta n} + 18\log(4/\delta)G^2\eta\right) + \frac{GD}{n^2} \\
&= \mathcal{O}\left(GD\left(\frac{1}{\sqrt{n}} + \frac{\sqrt{d}\log^{\frac{3}{2}}(1/\delta)}{\epsilon n}\right)\right),
\end{aligned}$$

where in the second inequality used $K = \lceil \log_2 n \rceil$, and the last inequality is due to $\eta = \frac{D}{G} \min\{\frac{\log(4/\delta)}{\sqrt{n}}, \frac{\epsilon}{12\log(4/\delta)\sqrt{2d\log(2.5/\delta)}}\}$. The desired excess generalization error bound is proved.

Finally, we investigate the gradient complexity argument. Since we run Algorithm 1 for $n_k$ at iteration $k$. Therefore, the total gradient complexity is $\mathcal{O}\left(\sum_{k=1}^{K} n_k\right) = \mathcal{O}(n\log(1/\delta))$. The proof is completed. $\qquad\square$

# E   Additional Results: Localization-Based Algorithm to Improve Theorem 1

In this section, we provide additional results on how to reduce the gradient complexity $\mathcal{O}(n^2)$ required in Theorem 1 to $\mathcal{O}(n)$ for nonsmooth problems. This improvement is attained by Algorithm 4 which is motivated by the iterative localization technique [13].

---

**Algorithm 4** Localized SGD for Pairwise Learning

---

1: **Inputs:** Dataset $S = \{\mathbf{z}_i : i = 1, \ldots, n\}$, parameter $\zeta$, initial point $\mathbf{w}_0$
2: Set $K = \lceil \log_2 n \rceil$ and divide $S$ into $K$ disjoint subsets $\{S_1, \cdots, S_K\}$ such that $|S_k| = n_k = 2^{-k}n$
3: **for** $k = 1$ to $K$ **do**
4:    Set $\zeta_k = 2^{-k}\zeta$
5:    Compute $\bar{\mathbf{w}}_k \in \mathcal{W}$ by Algorithm 1 with step sizes $\eta_j = \frac{\zeta_k n_k}{j+1}, j \in [T_k]$ and $T_k \asymp n_k$ iterations based on the objective $F_k$ where

$$F_k(\mathbf{w}; S_k) = \frac{1}{n_k(n_k-1)} \sum_{z,z' \in S_k : z \neq z'} f(\mathbf{w}; \mathbf{z}, \mathbf{z}') + \frac{1}{\zeta_k n_k}\|\mathbf{w} - \bar{\mathbf{w}}_{k-1}\|_2^2$$

6: **Outputs:** $\bar{\mathbf{w}}_K$

---

The next theorem shows that the empirical risk minimization can imply models with good excess generalization error by Algorithm 4.

**Theorem E.1.** *Let (A1) and (A3) hold true with $\alpha = 0$ and let $D$ be the diameter of $\mathcal{W}$. Let $\{\bar{\mathbf{w}}_k : k \in [K]\}$ be produced by Algorithm 4 with $\zeta = \frac{D}{G\sqrt{n}}$. Then we have the following excess generalization error bounds*

$$\mathbb{E}[F(\bar{\mathbf{w}}_K) - F(\mathbf{w}^*)] = \mathcal{O}\left(\frac{GD}{\sqrt{n}}\right)$$

*with gradient complexity $\mathcal{O}(n)$.*

We provide two technical lemmas before we present the proof of Theorem E.1.

**Lemma 8.** *Let (A1) and (A3) hold true with $\alpha = 0$ and let $\hat{\mathbf{w}}_k = \arg\min_{\mathbf{w}} F_k(\mathbf{w}; S_k)$, then*

$$\mathbb{E}[\|\bar{\mathbf{w}}_k - \hat{\mathbf{w}}_k\|_2^2] = \mathcal{O}\left(G^2\zeta_k^2 n_k\right).$$

*Proof.* Note that $F_k$ is $\alpha_k = \frac{2}{\zeta_k n_k}$-strongly convex, by the convergence of Algorithm 1 in Theorem 6 Part (b), we know that

$$\frac{\alpha_k}{2}\mathbb{E}[\|\bar{\mathbf{w}}_k - \hat{\mathbf{w}}_k\|_2^2] \leq \mathbb{E}[F_k(\bar{\mathbf{w}}_k; S_k) - F_k(\hat{\mathbf{w}}_k; S_k)] = \mathcal{O}\left(\frac{G^2}{\alpha_k n_k}\right)$$

which implies

$$\mathbb{E}[\|\bar{\mathbf{w}}_k - \hat{\mathbf{w}}_k\|_2^2] = \mathcal{O}\left(G^2\zeta_k^2 n_k\right).$$

The proof is completed. $\qquad\square$

**Lemma 9.** *Let (A1) and (A3) hold true with $\alpha = 0$. For any $\mathbf{w} \in \mathcal{W}$, we know that*

$$\mathbb{E}[F(\hat{\mathbf{w}}_k) - F(\mathbf{w})] \leq \frac{\mathbb{E}[\|\bar{\mathbf{w}}_{k-1} - \mathbf{w}\|_2^2]}{\zeta_k n_k} + 2G^2\zeta_k.$$

*Proof.* Let $r(\mathbf{w}; \mathbf{z}, \mathbf{z}') = f(\mathbf{w}, \mathbf{z}, \mathbf{z}') + \frac{1}{\zeta_k n_k} \|\mathbf{w} - \bar{\mathbf{w}}_{k-1}\|_2^2$, $R(\mathbf{w}) = \mathbb{E}_{\mathbf{z}, \mathbf{z}'}[r(\mathbf{w}; \mathbf{z}, \mathbf{z}')]$ and $\mathbf{w}_R^* = \arg\min_{\mathbf{w} \in \mathcal{W}} R(\mathbf{w})$. By the proof of Theorem 6 in Shalev-Shwartz et al. [33] , one has that

$$\mathbb{E}[F(\hat{\mathbf{w}}_k) + \frac{1}{\zeta_k n_k} \|\hat{\mathbf{w}}_k - \bar{\mathbf{w}}_{k-1}\|_2^2 - F(\mathbf{w}) - \frac{1}{\zeta_k n_k} \|\mathbf{w} - \bar{\mathbf{w}}_{k-1}\|_2^2]$$

$$= \mathbb{E}[R(\hat{\mathbf{w}}_k) - R(\mathbf{w})] \le \mathbb{E}[R(\hat{\mathbf{w}}_k) - R(\mathbf{w}_R^*)] \le 2G^2 \zeta_k,$$

which implies that

$$\mathbb{E}[F(\hat{\mathbf{w}}_k) - F(\mathbf{w})] \le 2G^2 \zeta_k - \frac{1}{\zeta_k n_k} \mathbb{E}[\|\hat{\mathbf{w}}_k - \bar{\mathbf{w}}_{k-1}\|_2^2] + \frac{1}{\zeta_k n_k} \mathbb{E}[\|\mathbf{w} - \bar{\mathbf{w}}_{k-1}\|_2^2]$$

$$\le 2G^2 \zeta_k + \frac{1}{\zeta_k n_k} \mathbb{E}[\|\mathbf{w} - \bar{\mathbf{w}}_{k-1}\|_2^2].$$

The proof is completed. $\qquad\square$

**Proof of Theorem E.1.** Let $\hat{\mathbf{w}}_0 = \mathbf{w}^*$, we have

$$\mathbb{E}[F(\bar{\mathbf{w}}_K)] - F(\mathbf{w}^*) = \sum_{k=1}^{K} \mathbb{E}[F(\hat{\mathbf{w}}_k) - F(\hat{\mathbf{w}}_{k-1})] + \mathbb{E}[F(\bar{\mathbf{w}}_K) - F(\hat{\mathbf{w}}_K)]. \qquad (\text{E.1})$$

For the first term we have

$$\sum_{k=1}^{K} \mathbb{E}[F(\hat{\mathbf{w}}_k) - F(\hat{\mathbf{w}}_{k-1})] \le \sum_{k=1}^{K} \left( \frac{\mathbb{E}[\|\bar{\mathbf{w}}_{k-1} - \hat{\mathbf{w}}_{k-1}\|_2^2]}{\zeta_k n_k} + 2G^2 \zeta_k \right)$$

$$= \mathcal{O}\left( \frac{D^2}{\zeta n} + \sum_{k=2}^{K} G^2 \zeta_k + \sum_{k=1}^{K} 2^{-k} G^2 \zeta \right)$$

$$= \mathcal{O}\left( \frac{D^2}{\zeta n} + G^2 \zeta \right) \qquad (\text{E.2})$$

where the first inequality is by Lemma 9, the second inequality is by Lemma 8 and $\zeta = \frac{D}{G\sqrt{n}}$. For the second term we have

$$\mathbb{E}[F(\bar{\mathbf{w}}_K) - F(\hat{\mathbf{w}}_K)] \le G \mathbb{E}[\|\bar{\mathbf{w}}_K - \hat{\mathbf{w}}_K\|_2] \le G \sqrt{\mathbb{E}[\|\bar{\mathbf{w}}_K - \hat{\mathbf{w}}_K\|_2^2]} = \mathcal{O}(G^2 \zeta_K \sqrt{n_K})$$

$$= \mathcal{O}\left( 2^{-2K} G^2 \zeta \sqrt{n} \right) = \mathcal{O}\left( G^2 \zeta \right) \qquad (\text{E.3})$$

where the first inequality is by $G$-Lipschitz continuity of $F$, the second inequality is by Jensen's inequality, the first identity is by Lemma 8 and the second identity is by $n_k = 2^{-k} n$.

Now putting (E.2) and (E.3) back to (E.1) and using $\zeta = \frac{D}{G\sqrt{n}}$, we derive

$$\mathbb{E}[F(\bar{\mathbf{w}}_K)] - F(\mathbf{w}^*) = \mathcal{O}\left( \frac{GD}{\sqrt{n}} \right).$$

Finally we investigate the gradient complexity. Since $F_k$ is $\frac{2}{\zeta_k n_k}$-strongly convex, by Theorem 6 Part (b), we need to choose $T_k \asymp n_k$ so that Lemma 8 holds. Therefore, in total, we require $\mathcal{O}\left( \sum_{k=1}^{K} n_k \right) = \mathcal{O}(n)$ gradient complexity, which yields the desired result. $\qquad\square$

# F  Additional Results: Differentially Private SGD for Pairwise Learning with Non-Smooth Losses

In this section, we propose a differentially private algorithm based on iterative localization [13] for nonsmooth pairwise learning problems. The algorithm is presented as follows.

We are now ready to present the privacy guarantee and utility bound of Algorithm 5 in the following theorem. The proof differs from the iterative localization algorithm in pointwise learning [13] since we employ our high probability convergence results for non-smooth losses in pairwise learning.

---

**Algorithm 5** Differentially Private Localized SGD for Pairwise Learning

---

1: **Inputs:** Dataset $S = \{\mathbf{z}_i : i \in [n]\}$, parameters $\epsilon, \delta$, and $\zeta$, initial points $\mathbf{w}_0$
2: Set $K = \lceil \log_2 n \rceil$ and divide $S$ into $K$ disjoint subsets $\{S_1, \cdots, S_K\}$ where $|S_k| = n_k = 2^{-k}n$.
3: **for** $k = 1$ to $K$ **do**
4:    Set $\zeta_k = 4^{-k}\zeta$
5:    Compute $\bar{\mathbf{w}}_k \in \mathcal{W}$ by Algorithm 1 with step sizes $\eta_j = \frac{\zeta_k n_k}{j+1}$ on objective $F_k$ such that with
     prob $1 - \delta$,
$$F_k(\bar{\mathbf{w}}_k; S_k) - \min_{\mathbf{w} \in \mathcal{W}} F_k(\mathbf{w}; S_k) \le G^2 \zeta_k / n_k$$
     where $F_k(\mathbf{w}; S_k) = \frac{1}{n_k(n_k-1)} \sum_{z,z' \in S_k : z \ne z'} f(\mathbf{w}; z, z') + \frac{1}{\zeta_k n_k} \|\mathbf{w} - \mathbf{w}_{k-1}\|_2^2$
6:    Set $\mathbf{w}_k = \bar{\mathbf{w}}_k + \mathbf{u}_k$ where $\mathbf{u}_k \sim \mathcal{N}(0, \alpha_k^2 I_d)$ with $\sigma_k = 4G\zeta_k\sqrt{\log(2.5/\delta)}/\epsilon$.
7: **Outputs:** $\mathbf{w}_K$

---

**Theorem F.1.** *Let (A1) and (A3) hold true with $\alpha = 0$ and let $D$ be the diameter of $\mathcal{W}$. Let $\{\mathbf{w}_k : k \in [K]\}$ be produced by Algorithm 5 with $\zeta = \frac{D}{G}\min\{\frac{4}{\sqrt{n}}, \frac{\epsilon}{4\sqrt{d\log(1/\delta)}}\}$. Then Algorithm 5 satisfies $(\epsilon, \delta)$-DP. Furthermore we have the following excess generalization error bounds*

$$\mathbb{E}[F(\mathbf{w}_K) - F(\mathbf{w}^*)] = \mathcal{O}\Big(GD\Big(\frac{1}{\sqrt{n}} + \frac{\sqrt{d\log(1/\delta)}}{\epsilon n}\Big)\Big)$$

*with no more than $\mathcal{O}(n^2 \log(1/\delta))$ stochastic gradient computations.*

**Proof of Theorem F.1.** We first consider the privacy guarantee of Algorithm 5. For any neighboring datasets $S = \{S_1, \dots, S_K\}$ and $S' = \{S'_1, \dots, S'_K\}$ differing by one example, where $S'$ follows the same partition as $S$, and $S_i \cap S_j = \emptyset$ if $i \ne j$. Let $\hat{\mathbf{w}}_k = \arg\min_{\mathbf{w}} F_k(\mathbf{w}; S_k)$ and $\hat{\mathbf{w}}'_k = \arg\min_{\mathbf{w}} F_k(\mathbf{w}; S'_k)$. We first investigate the $\ell_2$-sensitivity of $\hat{\mathbf{w}}_k$. Since $F_k$ is $\alpha_k = \frac{2}{\zeta_k n_k}$-strongly convex, by Theorem 6 in Shalev-Shwartz et al. [33] we have

$$\|\hat{\mathbf{w}}_k - \hat{\mathbf{w}}'_k\|_2 \le \frac{4G}{\alpha_k n_k} = 2G\zeta_k,$$

where $\bar{\mathbf{w}}'_k$ is the return from Line 5 in Algorithm 5 based on $F_k(\mathbf{w}; S'_k)$. By the strong convexity of $F_k$ again, we have with probability at least $1 - \delta$

$$\frac{\alpha_k}{2}\|\bar{\mathbf{w}}_k - \hat{\mathbf{w}}_k\|_2^2 \le F_k(\bar{\mathbf{w}}_k; S_k) - F_k(\hat{\mathbf{w}}_k; S_k) \le \frac{G^2 \zeta_k}{n_k}$$

which implies $\|\bar{\mathbf{w}}_k - \hat{\mathbf{w}}_k\|_2 \le G\zeta_k$. This further implies $\bar{\mathbf{w}}_k$ has $\ell_2$-sensitivity of $4G\zeta_k$ with probability $1 - \delta$. Therefore, by Lemma 5, each iteration $k$ of Algorithm 5 is $(\epsilon, \delta)$-DP. Since the partition of the dataset $S$ is disjoint, and each iteration $k$ of Algorithm 5 we only use one subset, thus the whole process will still be $(\epsilon, \delta)$-DP.

Next we investigate the utility bound of Algorithm 5. Firstly, for any fixed $\mathbf{w}$,

$$\mathbb{E}[F(\bar{\mathbf{w}}_k) - F(\mathbf{w})] = \mathbb{E}[F(\hat{\mathbf{w}}_k) - F(\mathbf{w})] + \mathbb{E}[F(\bar{\mathbf{w}}_k) - F(\hat{\mathbf{w}}_k)]$$
$$\le \frac{\mathbb{E}[\|\mathbf{w}_{k-1} - \mathbf{w}\|_2^2]}{\zeta_k n_k} + 3G^2 \zeta_k$$

where we used Lemma 9 and $\|\bar{\mathbf{w}}_k - \hat{\mathbf{w}}_k\|_2 \le G\zeta_k$. Denote $\bar{\mathbf{w}}_0 = \mathbf{w}^*$ and $\mathbf{u}_0 = \mathbf{w}_0 - \mathbf{w}^*$, we have

$$\mathbb{E}[F(\mathbf{w}_K) - F(\mathbf{w}^*)] = \sum_{k=1}^{K} \mathbb{E}[F(\bar{\mathbf{w}}_k) - F(\bar{\mathbf{w}}_{k-1})] + \mathbb{E}[F(\mathbf{w}_K) - F(\bar{\mathbf{w}}_K)]$$
$$\le \sum_{k=1}^{K} \Big(\frac{\mathbb{E}[\|\mathbf{u}_{k-1}\|_2^2]}{\zeta_k n_k} + 3G^2 \zeta_k\Big) + G\mathbb{E}[\|\mathbf{u}_K\|_2]. \tag{F.1}$$

Recall that by definition $\zeta \le \frac{D\epsilon}{4G\sqrt{d\log(2.5/\delta)}}$, so that for all $k \ge 0$, there holds

$$\mathbb{E}[\|\mathbf{u}_k\|_2^2] = d\sigma_k^2 = d\Big(\frac{4^{-k}G\zeta}{\epsilon}\Big)^2 \le 16^{-k}D^2.$$

Plugging the above estimate into (F.1) it follows

$$\mathbb{E}[F(\mathbf{w}_K) - F(\mathbf{w}^*)] \leq \sum_{k=1}^{K} 2^{-k} \Big( \frac{8D^2}{\zeta n} + 3G^2 \zeta \Big) + 4^{-K} GD$$

$$\leq \sum_{k=1}^{K} 2^{-k} GD \Big( \frac{8}{n} \max \Big\{ \sqrt{n}, \frac{\sqrt{d \log(1/\delta)}}{\epsilon} \Big\} + \frac{1}{2\sqrt{n}} \Big) + \frac{GD}{n^2}$$

$$\leq 9GD \Big( \frac{1}{\sqrt{n}} + \frac{\sqrt{d \log(1/\delta)}}{n\epsilon} \Big) + \frac{GD}{n^2}.$$

This yields the desired utility bound.

Finally, we investigate the gradient complexity argument. Since $F_k$ is $\frac{2}{\zeta_k n_k}$-strongly convex. We know from Theorem 7 Part (b), after $T_k \asymp n_k^2 \log(1/\delta)$ iterations, we have with probability $1 - \delta$

$$F_k(\bar{\mathbf{w}}_k; S_k) - \min_{\mathbf{w}} F_k(\mathbf{w}; S_k) = \mathcal{O}\Big( \frac{G^2 \zeta_k n_k \log(1/\delta)}{n_k^2 \log(1/\delta)} \Big) = \mathcal{O}\Big( \frac{G^2 \zeta_k}{n_k} \Big)$$

which satisfies the requirement at Line 5 of Algorithm 5. Therefore, in total the gradient complexity is of the form $\mathcal{O}\Big( \sum_{k=1}^{K} n_k^2 \log(1/\delta) \Big) = \mathcal{O}\big( n^2 \log(1/\delta) \big)$. The proof is completed. □

## G  Additional Experiments

In this section, we provide the experimental details and additional experiments to support our theoretical findings. The datasets we used are from LIBSVM website [9]. The statistics of the data is included in Table G.1. For data with multiple classes, we convert the first half of class numbers to be the positive class and the second half of class numbers to be the negative class.

Table G.1: Data Statistics. $n$ is the number of samples and $d$ is the number of features.

|   | diabtes | german | ijcnn1 | letter | mnist | usps |
|---|---------|--------|--------|--------|-------|------|
| n | 768 | 1,000 | 49,990 | 15,000 | 60,000 | 7,291 |
| d | 8 | 24 | 22 | 161 | 780 | 256 |

Table G.2: Average AUC score $\pm$ standard deviation across multiple datasets. Our best results are highlighted in bold.

| Algorithm | diabetes | german | ijcnn1 | letter | mnist | usps |
|-----------|----------|--------|--------|--------|-------|------|
| Our | **.831 ± .030** | .793 ± .021 | **.934 ± .002** | .810 ± .007 | **.932 ± .001** | **.926 ± .006** |
| SGD$_{pair}$ [27] | .830 ± .028 | .794 ± .023 | .934 ± .003 | .811 ± .008 | .932 ± .001 | .925 ± .006 |
| OLP [22] | .825 ± .028 | .787 ± .028 | .916 ± .003 | .808 ± .010 | .927 ± .003 | .917 ± .006 |
| OAM$_{gra}$ [48] | .828 ± .026 | .785 ± .029 | .930 ± .003 | .806 ± .008 | .898 ± .002 | .916 ± .005 |
| OLP-RS1 | .736 ± .074 | .630 ± .065 | .668 ± .026 | .683 ± .033 | .749 ± .045 | .737 ± .056 |
| OAM-RS1 | .737 ± .069 | .640 ± .058 | .677 ± .014 | .675 ± .050 | .685 ± .042 | .691 ± .059 |
| SPAUC [26] | .828 ± .031 | .799 ± .026 | .932 ± .002 | .809 ± .008 | .927 ± .002 | .923 ± .005 |

For each dataset, we have used 80% of the data for training and the remaining 20% for testing. The results are based on 25 runs of random shuffling. The generalization performance is reported using the average AUC score and standard deviation on the test data. To determine proper hyper parameters, we conduct 5-fold cross validation on the training sets: 1) for Algorithm 1 and SGD$_{pair}$, we select step sizes $\eta_t = \eta \in 10^{[-3:3]}$ and $\mathcal{W}$ diameter $D \in 10^{[-3:3]}$; 2) for OLP we select step sizes $\eta_t = \eta/\sqrt{t}$ where $\eta \in 10^{[-3:3]}$ and $\mathcal{W}$ diameter $D \in 10^{[-3:3]}$; 3) for OAM$_{gra}$ we select learning rate parameter $C \in 10^{[-3:3]}$; 4) for SPAUC we select step sizes $\eta_t = \eta/\sqrt{t}$ where $\eta \in 10^{[-3:3]}$.

Firstly, Table G.2 summarizes the generalization performance of different algorithms which contains more comparison results than Table 1. In particular, two additional results are added for comparison, i.e. OLP-RS1 and OAM-RS1 which denote the OLP [22] and OAM$_{gra}$ [48] with Reservoir sampling and the buffering set size $s = 1$, respectively. We can see that OLP-RS1 and OAM-RS1 are inferior to other algorithms. This inferior performance for OLP and OAM with a small buffering set was also observed in the experiments of [22, 48].

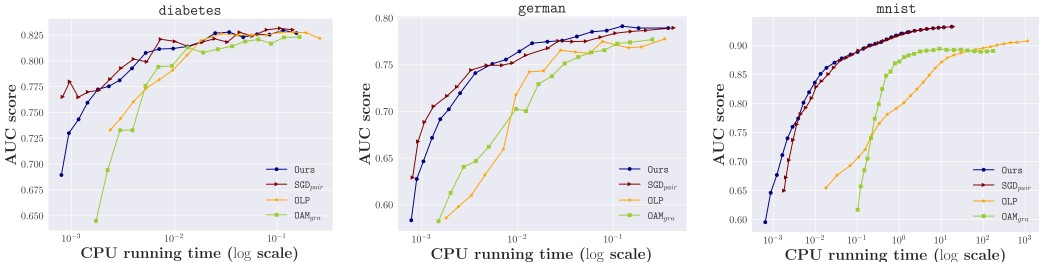

Figure G.1: More CPU running time against AUC score for the hinge loss

Secondly, we also report more plots on CPU running time against the AUC score on different datasets. Figure G.1 contains more convergence plots for the hinge loss. For a fair comparison of Algorithm 1 with SPAUC, the loss function is chosen as the least square loss for Algorithm 1, $SGD_{pair}$ and OLP. The results are shown in Figure G.2. We can see there that SPAUC performs very well among most of the datasets. However, this algorithm was designed very specifically for the AUC maximization problem with the least square loss while our algorithm can handle any loss functions and any pairwise learning problems. We can also observe that our algorithm and $SGD_{pair}$ converge in a similar CPU running time. In fact, Algorithm 1 is slightly faster than $SGD_{pair}$ when the number of samples gets larger. This is partly due to different sampling schemes in Algorithm 1 and $SGD_{pair}$. Indeed, at each iteration $SGD_{pair}$ picks a random pair of examples from $\binom{n}{2}$ pairs, while Algorithm 1 only needs to randomly pick one example from $n$ individual ones. Figure G.3 depicts the CPU times of these two sampling schemes versus the the number of examples $n$. We can see that, when the sample size $n$ increases, the sampling scheme used in $SGD_{pair}$ needs significantly more time than our algorithm.

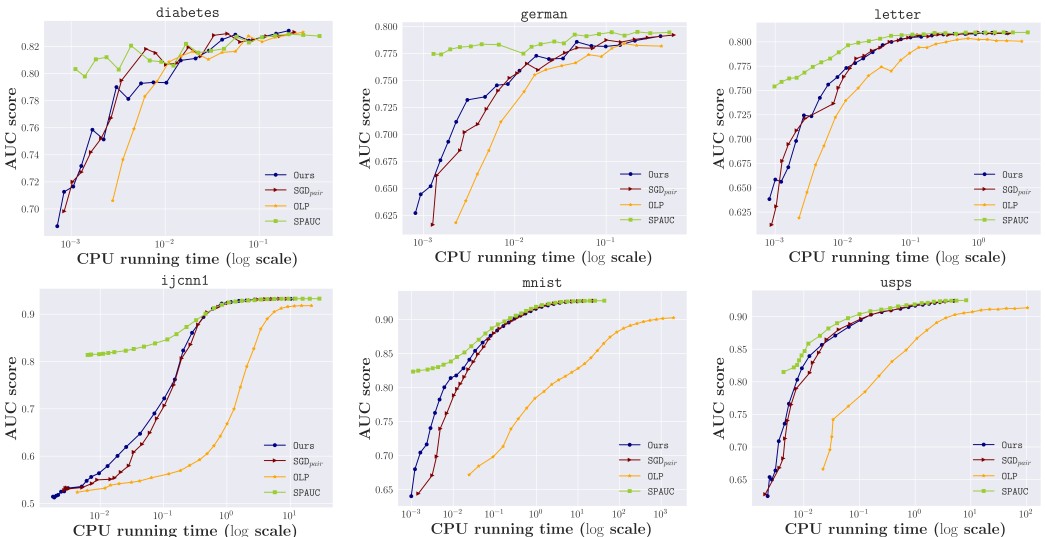

Figure G.2: AUC score against CPU running time for the square loss

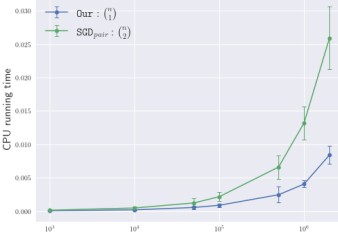

Figure G.3: CPU running time of different sampling schemes against the sample size $n$

Next, we investigate our Algorithm 1 in the non-convex setting. To this end, we use the logistic link function $\mathrm{logit}(t) = (1 + \exp(-t))^{-1}$ and then the square loss surrogate function $\ell(t) = (1 - t)^2$.

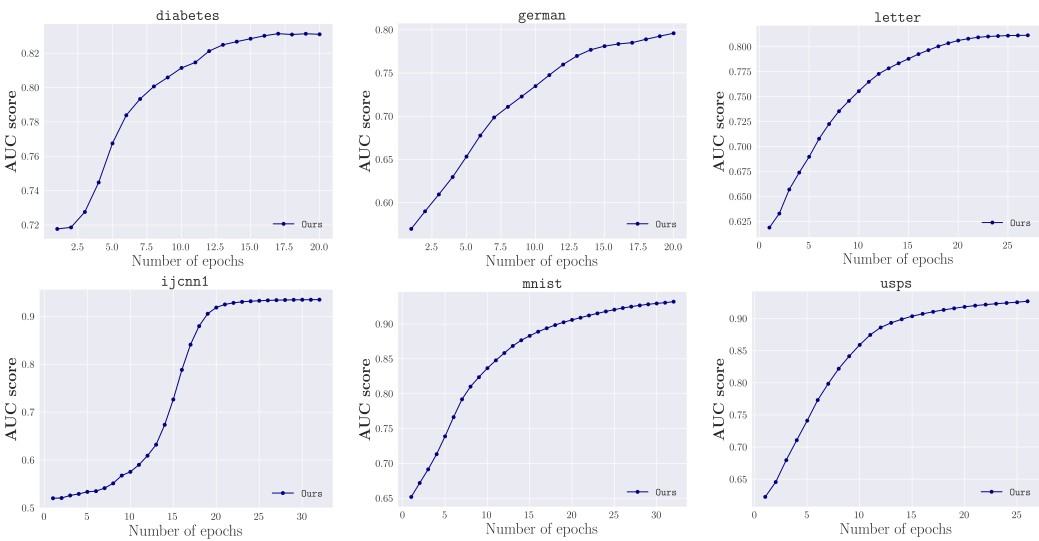

Figure G.4: Convergence of Algorithm 1 for the generalized linear model

That is, the loss function for the problem of AUC maximization becomes $f(\mathbf{w}; (\mathbf{x}, y), (\mathbf{x}', y')) = (1 - \text{logit}(\mathbf{w}^\top(\mathbf{x} - \mathbf{x}'))^2 \mathbb{I}_{[y=1 \wedge y'=-1]}$. Although $f$ is non-convex, it was shown that it satisfies the PL condition [14]. The results are reported in Figure G.4 which shows that Algorithm 1 also converges very quickly in this non-convex setting.

Finally, we compare our differentially private algorithm for AUC maximization (i.e. Algorithm 3) with the logistic loss $\ell(t) = \log(1 + \exp(-t))$ against the state-of-art algorithm DPEGD [42]. DPEGD used gradient descent and the localization technique to guarantee privacy. Algorithm 1 was used as non-private baseline, i.e. $\epsilon = 0$. Here, $\delta = \frac{1}{n}$ as suggested in the previous work [42]. We consider the effect of different privacy budget $\epsilon$'s against the generalization ability. The implementation across all algorithms is based on fixed training size 256. Average AUC scores over 25 times repeated experiments are listed in Table G.3 and G.4 for the datasets of diabetes and german, respectively. These results demonstrate Algorithm 3 achieves competitive performance with DPEGD using full gradient descent.

Table G.3: Average AUC $\pm$ standard deviation on diabetes. Non-Private result is $.813 \pm .016$.

| Algorithm | $\epsilon = 0.2$ | $\epsilon = 0.5$ | $\epsilon = 0.8$ | $\epsilon = 1.0$ | $\epsilon = 1.5$ | $\epsilon = 2.0$ |
|---|---|---|---|---|---|---|
| Our | $.690 \pm .094$ | $.751 \pm .028$ | $.771 \pm .016$ | $.783 \pm .024$ | $.784 \pm .018$ | $.789 \pm .018$ |
| DPEGD [42] | $.624 \pm .109$ | $.727 \pm .055$ | $.768 \pm .027$ | $.796 \pm .011$ | $.797 \pm .017$ | $.792 \pm .016$ |

Table G.4: Average AUC $\pm$ standard deviation on german. Non-Private result is $.763 \pm .016$.

| Algorithm | $\epsilon = 0.2$ | $\epsilon = 0.5$ | $\epsilon = 0.8$ | $\epsilon = 1.0$ | $\epsilon = 1.5$ | $\epsilon = 2.0$ |
|---|---|---|---|---|---|---|
| Our | $.614 \pm .035$ | $.672 \pm .064$ | $.721 \pm .024$ | $.725 \pm .032$ | $.747 \pm .019$ | $.749 \pm .021$ |
| DPEGD [42] | $.598 \pm .018$ | $.703 \pm .039$ | $.723 \pm .029$ | $.742 \pm .028$ | $.753 \pm .017$ | $.757 \pm .018$ |

We also report the CPU running times of Algorithm 3 and DPEGD. In this setting, we fix the privacy budget $\epsilon = 1$ and vary the training size $n$. The results are reported in Table G.5. These results shows that Algorithm 3 can arrive competitive performance with DPEGD with less CPU running time.

Table G.5: Average AUC score and average CPU running time $\pm$ standard deviation.

| Algorithm | | diabetes | | | german | | |
|---|---|---|---|---|---|---|---|
| | | $n = 100$ | $n = 200$ | $n = 300$ | $n = 100$ | $n = 200$ | $n = 300$ |
| Our | AUC | $.709 \pm .051$ | $.788 \pm .019$ | $.790 \pm .021$ | $.681 \pm .029$ | $.692 \pm .032$ | $.734 \pm .022$ |
| | Time | $.046 \pm .010$ | $.096 \pm .019$ | $.135 \pm .026$ | $.377 \pm .097$ | $.637 \pm .136$ | $.767 \pm .151$ |
| DPEGD [42] | AUC | $.705 \pm .070$ | $.772 \pm .017$ | $.777 \pm .023$ | $.687 \pm .033$ | $.700 \pm .038$ | $.755 \pm .019$ |
| | Time | $.421 \pm .067$ | $.885 \pm .158$ | $1.41 \pm .248$ | $1.00 \pm .185$ | $1.88 \pm .273$ | $2.57 \pm .421$ |