# OpenReview forum: "Simple Stochastic and Online Gradient Descent Algorithms for Pairwise Learning"
_NeurIPS.cc/2021/Conference — NeurIPS 2021 Poster_

### Official Review · Reviewer_7nrA · 2021-07-04

**Rating:** 6
**Confidence:** 3

**Summary:**

The paper proposed "stochastic gradient descent" (SGD) and "online gradient descent" (OGD) algorithms for pairwise learning problems. Stability results, optimization, and generalization error bounds for both convex and nonconvex as well as both smooth and nonsmooth problems are provided. The authors also develop differentially private SGD algorithms for pairwise learning. Empirical results are provided in a variety of real-world datasets.

**Limitations And Societal Impact:**

No.

See Main Review.

**Main Review:**

Originality: The novelty of the proposed pairwise model seems limited. This work is an extension of [20] by setting the buffer size of 1. The theoretical analysis is solid.

Quality: This paper provides worthy, but not surprising contributions to minor problems. The theoretical analysis is technically sound. However, I have several concerns. 1) It is unclear that why the proposed method performs better than [25] both in intuition and theory. 2) Experimental results are weak and improvements are not significant, especially for [25] and [24]. 3) Besides the running time, what about the AUC score w.r.t number of training examples?

Clarity: The paper is clearly written and provides enough information for an expert reader to reproduce its results.

Significance: The paper will have a low overall impact in practice. It is unclear whether it is a better way than previous work.

**Time Spent Reviewing:**

8

---

> ### Author Response · Authors · 2021-08-08
> **response to 7nrA**
>
>  First of all, we are grateful for your constructive suggestions and comments.
>
> **Q: The novelty of the proposed pairwise model seems limited. This work is an extension of [20] by setting the buffer size of 1. The theoretical analysis is solid.**
>
> **A:** In its format, the proposed algorithm is a special one of [20] with s=1 and FIFO strategy. That is why we call it simple SGD or OGD for pairwise learning.  Our main contribution is theoretical analysis of its stability, generalization and differential privacy for various cases (convex, nonconvex, smooth and non-smooth setting) where the analysis in [20] can not handle as discussed in the introduction.  We therefore give an affirmative answer to the open question proposed in [20].
>
> In particular, our analysis technique is novel and is fundamentally different from [20]. As mentioned in the discussion in Remark 4,  [20] removes the coupling by a uniform convergence argument (Rademacher Complexity) while we propose novel techniques tailored for Algorithm 1 and Algorithm 2 to handle the coupling. With the help of the developed technique, we are able to answer an open question raised in [20] regarding the generalization of learning with a buffering set of a very small fixed size. Moreover, the differential privacy results significantly improve the existing work which has not been addressed in [20].
>
> **Q: It is unclear why the proposed method performs better than [25] both in intuition and theory.**
>
> **A:** As we discussed in detail in Remark 2, there are three differences from [25]. Firstly, our Algorithm 2 can handle the important online setting  (i.e. streaming data where examples arrive in a real time manner), while [25] can not as it randomly selects one pair of examples from all n(n-1)/2 pairs (which requires the data is available before running the algorithm).  Secondly, from the theoretical aspect, our algorithms involve a biased gradient at each iteration while [25] used an unbiased one.  Hence, our analysis needs novel techniques to handle the biased gradient and is more tricky. Thirdly, we provided generalization results for nonsmooth, nonconvex losses and also used Algorithm 1 to develop novel differentially private pairwise learning algorithms while [25] focused on the smooth convex losses and did not address differential privacy.
>
> **Q:  Experimental results are weak and improvements are not significant, especially for [25] and [24].**
>
> **A:** We want to emphasize that the main contribution is theoretical study of stability, generalization and differential privacy of simple SGD or ODG for pairwise learning which resolves an open question with very small batch size (s=1).  The preliminary experiments are designed to validate the theoretical findings.
>
> Our algorithm is general for pairwise learning problems with general losses while  [24] is tailored for the AUC maximization problem with the square loss function.  As for the difference from [25], please see Remark 2 in the paper and also our response to the previous question.
>
> **Q: Besides the running time, what about the AUC score w.r.t number of training examples**
>
> **A:** Thank you for the constructive suggestion. We will report AUC scores w.r.t. the number of training examples in the revised version.

---

> > ### Comment · Reviewer_7nrA · 2021-08-20
> > **Response to authors.**
> >
> > Thank you for the response to the comments. Some of my concerns are addressed. I will increase the score to be 6 and would like to see more experiments in the further version.

---

### Official Review · Reviewer_PEqh · 2021-07-05

**Rating:** 7
**Confidence:** 3

**Summary:**

This paper proposes simple offline and online algorithms of pairwise learning which share the same idea of updating parameters based on only the current and the last seen data points. Then follow a series of generalization analysis on these algorithms under various loss function conditions. Finally simple but crucial experimental results show the feasibility of the proposed methods.

**Limitations And Societal Impact:**

Not applied.

**Main Review:**

Originality: In pairwise learning, this paper proposes to update parameters using the current data point and only the last seen data point, in both offline and online setting. Although this causes the problem of the gradient update no longer an unbiased estimate of the true gradient, authors propose two interesting methods to alleviate this: composing indicator functions and using the difference of $\mathcal{O}(\eta_{t-1})$ between the evaluations before and after the parameter update.

Quality: Claims are theoretically sound and are well supported theoretically and empirically.

Clarity: This paper is very well written and easy to follow.

Significance: The proposed method reduce the gradient complexity to a surprising $\mathcal{O}(1)$ which significantly improves its practicability in real world.


**Time Spent Reviewing:**

2

---

> ### Author Response · Authors · 2021-08-08
> **response to reviewer PEqh**
>
> Thank you for your encouraging words and comments.

---

### Official Review · Reviewer_nSDq · 2021-07-16

**Rating:** 7
**Confidence:** 2

**Summary:**

The authors consider the pairwise learning problem in which the loss functions depends on two datapoints and not only one as in standard supervised learning. In order to address this problem, state-of-the-art methods use stochastic and online gradient descent algorithm, pairing the current datapoint with a buffering set of previous datapoints with an appropriate size. In this work, the authors propose as alternative to pair the current instance with only the previous one when building a gradient direction for the update step. This is obviously advantageous both for the storage and the computational complexity. The authors also present new stability results, optimization, and generalization error bounds for their method for convex/non-convex and smooth/non-smooth problems. The results they find is competitive to the rates for standard supervised learning. Finally, they also extend their pairwise algorithms and the corresponding theoretical analysis to a differentially private context improving state-of-the-art results.

**Ethical Concerns:**

no ethical issue in my opinion

**Limitations And Societal Impact:**

yes

**Main Review:**

The paper presents an interesting problem and it is well written and easy to follow. The results seem to be reasonable.

The main contribution of the paper is for sure the adaptation of stochastic and online learning algorithm to pairwise learning by keeping the same computational cost and convergence rate as in the (singleton) standard supervised learning setting. This is technically done by using as descent direction the subgradient of the loss function evaluated at the current point and the previous one. From the theoretical point of view, in order to provide guarantees for such a method, the authors use standard tools from stability theory, but they also need to introduce a novel technique allowing them to decouple the dependency of models and the previous instance in both the optimization and generalization analysis. It would be nice and important to describe in the main body the high-level idea used during this step.

The authors say that pairwise learning instantiates many important machine learning tasks such as bipartite ranking and metric learning. I would like to see a more extensive justification about the possible applications of pairwise learning, explaining to the reader why pairwise learning could be interesting for the machine learning community.

The experiments seem to be exhaustive. The authors test the performance of their algorithms on a AUC maximization problem by comparing the proposed algorithm to four existing algorithms for pairwise learning in terms of generalization and CPU running time on several datasets from the LIBSVM website.

**Time Spent Reviewing:**

48

---

> ### Author Response · Authors · 2021-08-08
> **response to reviewer nSDq**
>
>
> First of all, thank you very much for your constructive comments and suggestions.
>
> **Q: It would be nice and important to describe in the main body the high-level idea used during this step.**
>
> **A:** Thank you for your suggestion. We provided some discussion on our technique of decoupling in Remark 4, for both generalization analysis and optimization analysis.  Following your suggestion, we will describe the high-level ideas in more detail in the revised version.
>
> **Q: I would like to see a more extensive justification about the possible applications of pairwise learning, explaining to the reader why pairwise learning could be interesting for the machine learning community.**
>
> **A:** Thank you for your constructive suggestion. Pairwise learning covers many machine learning tasks such as AUC maximization, metric learning and minimum error entropy principle. We will include more concrete examples of pairwise learning applications in the revised version.

---

> > ### Comment · Reviewer_nSDq · 2021-08-11
> > **After Rebuttal**
> >
> > Thanks to the authors for the reply. The authors addressed my questions/concerns saying that they will include a high-level idea of their statements and a deeper discussion about the applications of pairwise learning in the final version. Because of this, I decide to maintain my mark as it is.

---

### Official Review · Reviewer_7Peg · 2021-07-18

**Rating:** 8
**Confidence:** 2

**Summary:**

This paper introduces a simple algorithm for online and offline pairwise learning, pairing the current sample only with the previous one. This is a more challenging setting than previously studied, as the gradient estimator is biased (the current weights and previous sample are coupled). Using novel techniques, this work derives an optimal excess generalization bound for this algorithm. Furthermore, the authors introduce a differentially private version of their algorithm, and derive the corresponding utility bound.
The theoretical results are validated by experiments on AUC maximization on a number of classic datasets, showing excellent results both in average AUC score and CPU running time when compared to the state-of-the-art.

**Limitations And Societal Impact:**

There is currently only a limited discussion of some limitations in the supplementary. This is one point I feel the authors could improve, as explained above in Q.1.

**Main Review:**

*For ease of answering, I have annotated my various points with O.1/Q.1/etc...*

My review will be quite short, as this is a very good paper. The idea proposed is conceptually very simple, but requires significant theoretical work to derive useful bounds. The paper is well written, and the authors do a great job at outlining their mathematical demonstrations. The experimental validation is thorough, with extensive further experiments in the supplementary.

Accordingly, this is a **clear accept (8)**. I only have two small points below on the style & clarity of the tables, and on the lack of a discussion around limitations & future work (especially in regard of Figure G.2 of the supplementary).

### Originality
This paper is well placed in the relevant literature, and the authors do a good job in section 2 at outlining how their work improves on the state-of-the-art. Although the algorithm proposed may look simple, all the complexity lies in providing useful bounds. Previous literature with buffering set of arbitrary size had only succeeded in providing bounds that are meaningful for very large batches.

### Quality
The theoretical part of this work is very solid, with a well-structured and excellent analysis. The experimental validation is equally good, comparing to a large number of methods on diverse datasets.

However, on light of Figure G.2 of the supplementary, I do have a (small) point of criticism.
**Q.1**: This work is very complete and leaves no question "hanging", but the authors could still make an effort to discuss potential limitations. It would be interesting to at least mention that methods such as SPAUC, designed for very specific losses and problems, can still be competitive (as seen in Figure G.2, and as sparsely discussed on l775 of the supplementary).

### Clarity
This paper is well written, and the theoretical analysis is easy to follow from the outline provided.

One small point I have to make is regarding the design used for the tables presenting results (Table 1, and in Section G of the supplementary).
**C.1**: The tables do not follow the style guidelines provided in the NeurIPS template. They use vertical rules, and too many horizontal rules. Given their density, this makes them hard to parse. Moreover, in each column, highlighting the best result in bold would make them more readable at a glance.

### Significance
This work clearly builds upon the state-of-the-art, and provides new useful bounds that are a clear improvement over previous work. This is a very interesting result, that answers a question previously left open. The techniques used by the authors to achieve this result may also be of interest to others in the field.

**Time Spent Reviewing:**

4

---

> ### Author Response · Authors · 2021-08-08
> **response to reviewer 7Peg**
>
> First of all, thank you very much for your encouraging words and very constructive suggestions.
>
> **Q: This work is very complete and leaves no question "hanging", but the authors could still make an effort to discuss potential limitations. It would be interesting to at least mention that methods such as SPAUC, designed for very specific losses and problems, can still be competitive (as seen in Figure G.2, and as sparsely discussed on l775 of the supplementary).**
>
> **A:** Great point! Indeed, SPAUC focused on the least square loss and AUC maximization which enjoys competitive generalization performance, with an even faster convergence rate sometimes, as argued in the paper. This also coincides with the experiments. One potential limitation of our proof techniques is that it remains an open question to us whether generalization and convergence still hold true if the current example $z_{i_t}$ is paired with one arbitrary previous example (e.g.,  $z_{i_t}$ is paired with $z_{i_1}$ or $z_{i_2})$.  Other future work would be systematic extension of our algorithms using other acceleration schemes such as momentum. We will include these useful discussions in the revised version.
>
> **Q: The tables do not follow the style guidelines provided in the NeurIPS template**
>
> **A:** Thank you for careful reading. We will address this issue in the revised version.

---

> > ### Comment · Reviewer_7Peg · 2021-08-17
> > **Thanks for addressing my comments.**
> >
> > Thanks to the authors for taking the time to address my comments.
> >
> > I maintain my rating as a **clear accept (8)**.

---

### Decision · Program_Chairs · 2021-09-28

**Decision:**

Accept (Poster)

**Comment:**

All the reviewers agree upon the quality of the paper and support its acceptance. There is no argument to go against this consensus and I recommend the acceptance of this paper.

We count on the authors to take into account the small fixes recommended by the reviewers.



**Consistency Experiment:**

NeurIPS has a long history of experimentation. In 2014, NeurIPS ran an experiment in which 10% of submissions were reviewed by two independent committees to quantify the randomness in the review process. This year, we repeated a variant of this experiment to see how the quality of the review process has changed over time.  This paper was part of the experiment and was therefore assigned to two committees (consisting of reviewers, an Area Chair, and a Senior Area Chair) that reached independent decisions.  If both committees made the same recommendation, this recommendation was followed. If a single committee recommended acceptance, the paper was accepted (with the exception of a few cases in which the other committee identified what we considered a fatal flaw, e.g., an error in a key result).

Both committees reached the same decision: **Accept (Poster)**

The other committee assigned to the paper recommended **Accept (Poster)**.  You can find the other set of reviews, along with any follow up discussion with the authors here:
https://openreview.net/forum?id=VXraeNhj4zI